  

# Inhibitory and synergistic effects of volatile organic compounds from bat caves against *Pseudogymnoascus destructans in vitro*

Zihao Huang,[1] Shaopeng Sun,[1] Yihang Li,[1] Zizhen Wei,[1] Mingqi Shen,[1] Jiaqi Lu,[1] Keping Sun,[2] Zhongle Li,[1,3] Jiang Feng[1,3]

**ABSTRACT** Fungi are ubiquitous in natural ecosystems, and environmental reservoirs such as bat hibernacula can harbor fungal pathogens and shape disease dynamics. Beyond serving as pathogen reservoirs, these environments may also contain volatile organic compounds (VOCs) with antifungal properties that help a host resist infection. Studies have shown that various VOCs from bat caves significantly inhibit the growth of *Pseudogymnoascus destructans*, the pathogen responsible for white-nose syndrome (WNS), although the underlying mechanisms remain unclear. This study investigates two VOCs isolated from bat cave environments—isovaleric acid (IVA) and ethyl methyl carbonate (EMC)—to evaluate their single-agent and combination activities against *P. destructans in vitro* and to explore the underlying mechanisms. The results show that both IVA and EMC significantly inhibit mycelial growth in a dose-dependent manner and exhibit synergistic antifungal effects. Physiological and biochemical analyses revealed that VOC treatment disrupts cell wall and membrane integrity, induces apoptosis, elevates reactive oxygen species levels, and causes DNA damage. Concentrations of adenosine triphosphate, malondialdehyde, ergosterol, and NADPH also increased significantly. Transcriptomic and metabolomic analyses showed disruption of the mycelial structure, modulation of virulence-associated pathways, induction of oxidative stress and apoptosis, and interference with purine metabolism, cAMP signaling, and energy metabolism. Notably, combined IVA–EMC treatment enhanced DNA damage and suppressed heat shock protein expression, effectively inhibiting *P. destructans* growth. Taken together, our study elucidates the antifungal potential of environmental VOCs and offers new insights and application prospects for preventing and controlling WNS.

**IMPORTANCE** White-nose syndrome has devastated bat populations across North America, yet effective control measures remain limited. This study highlights the potential of naturally occurring volatile organic compounds from bat cave environments as antifungal agents against *Pseudogymnoascus destructans in vitro*. By uncovering the physiological and molecular mechanisms of the action of isovaleric acid and ethyl methyl carbonate, individually and in combination, this work paves the way for novel, environmentally derived strategies for managing white-nose syndrome and fungal pathogens more broadly.

**KEYWORDS** environmental reservoirs, *Pseudogymnoascus destructans*, volatile organic compounds (VOCs), multi-omics, antifungal effects

E nvironmental reservoirs are host-free sites (e.g., cave walls, sediments, and guano) that maintain pathogen viability and seed new infections, constituting major sources of disease (e.g., chytridiomycosis, white-nose syndrome [WNS], and plague) that can influence pathogen evolution and potentially exacerbate host extinction risks (1). WNS is a psychrophilic fungal infection in bats that is caused by *Pseudogymnoascus destructans*.

**Peer Reviewer** Christine E. Salomon, University of Minnesota, Minneapolis, Minnesota, USA

Address correspondence to Zhongle Li, lzy1514316@126.com, or Jiang Feng, fengj@nenu.edu.cn.

The authors declare no conflict of interest.

See the funding table on p. 17.

10.1128/msystems.00903-25   **1**

This infection has caused bat populations to decline by over 90% in at least three bat species in North America since its discovery in New York in 2006 (2). In North America, environmental reservoirs of *P. destructans* are considered a major threat to bat populations as they serve as sources of infection for bats returning each hibernation season (3). In contrast, Chinese bat populations remain unaffected by WNS, and it has been found that both environmental and host-associated *P. destructans* loads are significantly lower than those in North America (2, 4). Ultimately, several previous studies suggest that the immune function and skin microbiota of Chinese bats contribute to their resistance to *P. destructans* infection (5, 6). This pattern is likely the result of a long coevolutionary history between bat hosts and *P. destructans*, which may have favored host tolerance or resistance (7). However, whether environmental reservoirs influence disease outcomes remains unclear.

The environment serves not only as a reservoir for pathogenic fungi but also as a key site for microbial interactions. In recent years, volatile organic compounds (VOCs) produced by microorganisms have attracted significant research interest (8). In particular, these small, volatile metabolites can diffuse through air and exhibit antifungal properties (9). VOCs originate from diverse sources and exhibit a wide range of biological activities. In complex ecosystems like bat caves, VOC-based biocontrol strategies offer distinct advantages due to their rapid dispersion and direct action on pathogenic fungi (10).

Previous studies have reported that various VOCs derived from bat cave environments, including nonanal, pentyl ester, propanoic acid, and 1-pentanol, have shown strong inhibitory effects against *P. destructans* (11, 12). In plant pathogen control, VOCs primarily act by disrupting cell wall and membrane integrity, which induces oxidative stress and triggers apoptosis (13–15). Similarly, trans-2-hexenal reduces *P. destructans* virulence by downregulating multiple pathogenicity-related genes (16). Additionally, previous transcriptomic analysis shows that *P. destructans* upregulates genes involved in heat shock response, cell wall remodeling, and trace element metabolism during infection, which suggests that these processes may be key targets for VOC activity (17). However, the specific mechanisms by which VOCs affect *P. destructans* remain unclear and warrant further investigation. In natural environments, multiple VOCs often coexist, and the antifungal efficacy of individual compounds may be limited (18). Taken together, it is important to explore whether synergistic interactions among VOCs enhance the antifungal activity.

In our previous research, we identified several VOCs with antifungal activity against *P. destructans* from soil and air samples collected at three *P. destructans*-positive bat hibernacula in northeastern China. Among these, isovaleric acid (IVA) and ethyl methyl carbonate (EMC) were consistently detected across all three sites. Each compound accounted for more than 1% of the total relative abundance, which indicates that both compounds are ubiquitously expressed and relatively enriched in these environments. Previous studies have also confirmed that acid and ester compounds exhibit strong activity against *P. destructans* (12). To explore potential synergistic effects, we systematically investigate the individual and combined effects of IVA and EMC on *P. destructans* using physiological, biochemical, transcriptomic, and metabolomic analyses. Altogether, this work advances our understanding of interactions between environmental reservoirs and pathogenic fungi and provides a theoretical basis for VOC-based control strategies against WNS.

## MATERIALS AND METHODS

### Materials and chemicals

*P. destructans* strain JHCN111a was stored as spore stocks in 30% (vol/vol) glycerol at −80°C (4). IVA (CAS:503-74-2, purity 99%) and EMC (CAS:623-53-0, purity 98%) were purchased from Macklin (Shanghai, China) and stored at 28°C. The fungal fluorescence stain kit (CFW method) was obtained from Solarbio Science and Technology Co., Ltd. (Beijing, China). Reactive oxygen species (ROS) assay kit (DCFH-DA), Annexin V-FITC

apoptosis detection kit, and 4',6-diamidino-2-phenylindole (DAPI) staining solution were purchased from Beyotime Biotechnology (Shanghai, China). The NADPH content assay kit was obtained from Nanjing Jiancheng Bioengineering Institute (Nanjing, China). Malondialdehyde (MDA) and adenosine triphosphate (ATP) content assay kits were sourced from Box Biotechnology Co., Ltd. (Beijing, China).

## Revival of *P. destructans* and vapor-phase exposure to VOCs

For revival, frozen *P. destructans* stocks were thawed and inoculated onto Sabouraud dextrose agar (SDA) and then incubated at 13°C and 85% relative humidity for 14 days. After incubation, 10 mL of 1× phosphate-buffered saline (PBS) with Tween 20 was added, and conidia were collected using a sterile loop (5). The suspension was adjusted to $2 \times 10^5$ conidia/mL with sterile water, counted using a hemocytometer, and used to evenly spread-inoculate SDA plates with a sterile spreader (100 µL per 90-mm plate). Plates were allowed to air-dry (10–15 min) before exposure.

For vapor-phase assays, the inoculated plates were inverted so that the lid served as the base. Sterile antibiotic disks (6 mm; BKMAN, China) were placed on the center of the inner surface of the inverted Petri dish lid. IVA and EMC were applied neatly to the disks at preset loading volumes to achieve target nominal headspace concentrations. Nominal concentration was defined as µL of compound per mL of headspace air (µL/mL) and calculated as follows: $C_{nom} = V_{vocs}/V_{headspace}$, where $V_{headspace}$ is the air volume of the sealed plate (total internal volume minus agar volume). Headspace volume was calculated from the plate geometry (90 mm diameter; 20 mm internal height) and the measured agar depth. After VOC loading, the plates remained in the inverted position and were sealed with Parafilm. They were incubated at 13°C and 85% relative humidity for 14 days. Because the plates were maintained in an inverted position, gravity ensured that the disks remained stationary on the lids throughout incubation without the need for adhesives. Controls consisted of sterile antibiotic disks loaded with 95% ethanol at the maximum solvent volume used for each treatment (vehicle control).

Digital images were analyzed with Ilastik v1.4.1rc2 (19) and ImageJ (20) to quantify the mycelial coverage area. The inhibition rate was computed as inhibition (%) = 1 − ($A_t/A_c$) × 100, where $A_t$ and $A_c$ are the mycelial areas of control and treatment plates, respectively (21). The minimum inhibitory concentration (MIC) was defined as the concentration of VOCs that completely inhibited the *P. destructans* mycelial growth, whereas the half-maximal inhibitory concentration ($IC_{50}$) was defined as the concentration that inhibited growth by 50%.

## Determination of synergistic interactions

A vapor-phase checkerboard assay was used to evaluate interactions between the two compounds (22). Based on single-agent MICs determined in prior gradient tests, six dose levels (1/16×, 1/8×, 1/4×, 1/2×, 1×, and 2×) were tested in a 6 × 6 matrix. For each combination, two sterile antibiotic disks (6 mm) were affixed to the underside of the Petri dish lid and separately loaded with IVA and EMC. After loading, the plates were left undisturbed for 30–60 s to allow solvent evaporation and were then sealed with Parafilm for incubation. Each test included vehicle and growth controls; the vehicle control used anhydrous ethanol as the loading solvent, with its composition and volume matched to the maximum solvent volume used within the matrix. All plates were incubated at 13°C and 85% relative humidity for 14 days. The fractional inhibitory concentration index (FICI) was calculated only for combinations that reached the MIC: FIC_IVA = C_IVA / MIC_IVA, FIC_EMC = C_EMC / MIC_EMC, and FICI = FIC_IVA + FIC_EMC. Here, MIC_IVA and MIC_EMC denote the single-agent MICs of IVA and EMC, and C_IVA and C_EMC are the concentrations of IVA and EMC that achieved complete inhibition under combination conditions, respectively. FICI thresholds were interpreted as follows: ≤0.5, synergy; > 0.5–1.0, additivity; 1.0–4.0, indifference; ≥4.0, antagonism. Combinations that did not reach the MIC were recorded as NA and excluded from FICI calculations. Each matrix was performed in three independent biological replicates.

## Cell wall and membrane damage assessments

Cell wall and cell membrane integrity were assessed following the method described by 23, with minor modifications. After 14-day exposure to VOCs at the $IC_{50}$, 20 mg of wet mycelial per sample was collected from treatment and control plates and washed twice in PBS. For cell wall staining, a calcofluor white (CFW) staining solution was prepared by diluting the supplier's stock 1:1 (vol/vol) with PBS; 1 mL per sample was applied and incubated for 30 min in the dark. For membrane integrity, a propidium iodide (PI) staining solution was prepared by mixing 10 µL PI stock to 190 µL PBS (1:20, vol/vol). The 200 µL solution was applied per sample and incubated for 30 min at room temperature in the dark. The integrity of the mycelial cell wall and membrane was examined using confocal laser scanning microscopy (CLSM, Leica, Germany).

MDA content in *P. destructans* mycelia was measured according to the method of 24, with slight modifications. For each replicate, 0.1 g of wet mycelia was collected and homogenized on ice in the kit-supplied extraction buffer (1:10, wt/vol). The homogenate was clarified by centrifugation at $8,000 \times g$ for 10 min at 4°C. The supernatant was collected, and MDA levels were quantified following the manufacturer's protocol. Ergosterol content was measured using the method described by 25). For each replicate, 0.1 g of wet mycelia was flash-frozen in liquid nitrogen, ground to a fine powder, and transferred to a 15 mL tube. A volume of 5 mL of 25% potassium hydroxide in ethanol was then added. Samples were vortexed for 2 min before incubation in an 85°C water bath for 4 h to facilitate saponification. After incubation, 2 mL of sterile distilled water and 5 mL of heptane were added sequentially. The mixture was left undisturbed for 1 h to allow phase separation. The upper heptane layer was collected, and absorbance was measured at 230 nm and 282 nm using a microplate reader (Thermo Scientific, USA). Ergosterol content was calculated using the following equations:

Ergosterol (%) + dehydroergosterol (%) = $(A_{282}/290)$/weight

Dehydroergosterol (%) = $(A_{230}/518)$/weight

Here, 290 and 518 are the extinction coefficients (E values) of ergosterol and dehydroergosterol, respectively.

## Detection of cellular energy metabolism and oxidative stress

Mycelial samples were treated with VOCs at the $IC_{50}$ concentrations and collected after 14 days. Intracellular ATP and NADPH levels after VOC treatments were quantified using commercial assay kits following the manufacturer's instructions. Briefly, 0.1 g of wet mycelia was homogenized on ice in the kit-supplied extraction buffer (1:10, wt/vol) and then clarified by centrifugation at $12,000 \times g$ for 10 min at 4°C; the supernatants were used for assays. ROS levels were assessed with an ROS assay kit (DCFH-DA). Washed mycelia (20 mg) were resuspended in 500 µL PBS containing 10 µM 2′,7′-dichlorodihydrofluorescein diacetate (DCFH-DA) and incubated at 37°C for 20 min with gentle inversion every 3–5 min. After incubation, ROS accumulation in mycelia was observed using CLSM.

## Detection of cellular DNA damage and apoptosis

Mycelial samples were treated with VOCs at the $IC_{50}$ concentrations and collected after 14 days. Samples were then fixed in 70% ethanol (500 µL per 20 mg wet biomass) for 30 min to preserve the cellular morphology and structure (26). After fixation, samples were washed twice with PBS to remove residual ethanol. For nuclear staining, mycelia were resuspended in 500 µL PBS containing DAPI (1 µg/mL), incubated for 30 min in the dark, washed twice with PBS, and imaged by CLSM.

For apoptosis assays, washed mycelia were resuspended in Annexin V binding buffer. To 200 µL of the suspension, annexin V–fluorescein isothiocyanate (5 µL) and PI (10 µL) were added, gently mixed, and incubated for 20 min at 25°C in the dark. Samples were then washed once with binding buffer and imaged by CLSM.

## Transcriptomic analysis

A total of three individual mycelial samples were processed with VOCs at the $IC_{50}$ concentrations and collected after 14 days. Total RNA was extracted using TRIzol reagent (Invitrogen, USA) (27). RNA concentration and integrity were assessed using a Qubit 2.0 fluorometer (Invitrogen, USA) and an Agilent 2100 Bioanalyzer (Agilent Technologies, USA) (28). High-quality RNA was used for library construction, followed by paired-end sequencing on the DNBSEQ-T7 platform (BGI, Shenzhen, China).

Raw reads were quality-checked using FastQC (29), and adapter sequences and low-quality bases were removed using Trimmomatic (30). Clean reads were aligned to the reference genome using Hisat2 (31) and gene expression levels were quantified as transcripts per million using StringTie (32). Principal component analysis (PCA) was conducted in R using the Vegan package to evaluate sample clustering (33). Weighted gene co-expression network analysis (WGCNA) was also conducted using the WGCNA package (34). Differentially expressed genes (DEGs) were identified using DESeq2 with thresholds of $|\log_2$ fold change$| \geq 1$ and a false discovery rate $< 0.05$ (35). Kyoto Encyclopedia of Genes and Genomes (KEGG) pathway enrichment analysis was also conducted using ClusterProfiler based on KEGG Orthology (KO) annotations (36). Analyses were run against the full KO reference database, with the gene universe defined as all expressed genes.

## Real-time quantitative PCR (RT-qPCR) analysis

To validate our RNA-seq results, 12 key genes were selected from the transcriptome for RT-qPCR analysis. Total RNA was reverse-transcribed into cDNA using the StarScript Pro All-in-one RT Mix with gDNA Remover (GeneStar, China). qPCR was performed using the 2× RealStar Fast SYBR qPCR Mix (Low ROX) (GeneStar, China) in a total reaction volume of 50 µL. The reactions were run on a QuantStudio 3 System (Thermo Scientific, USA) with the following cycling conditions: pre-denaturation at 95°C for 2 min, followed by 40 cycles of 95°C for 10 s and 60°C for 30 s. EFG1 was used as the reference gene, and relative gene expression was calculated using the $2^{-\Delta\Delta Ct}$ method (37). Specific primer sequences for all target genes are listed in Table S1.

## Metabolomic analysis

Six mycelial samples were treated with VOCs at the $IC_{50}$ concentrations and collected after 14 days. Metabolites were also extracted following the method described by 38. Briefly, 200 mg of mycelial biomass was mixed with 1 mL of a pre-chilled methanol, acetonitrile, and water solution (2:2:1, vol/vol) that was then ultrasonicated at 4°C for 30 min and incubated at −20°C for 10 min. The samples were centrifuged at 14,000 × $g$ for 20 min at 4°C, and the supernatant was vacuum-dried. The dried extracts were reconstituted in 100 µL of an acetonitrile–water solution, vortexed thoroughly, and centrifuged at 14,000 × $g$ for 15 min at 4°C. The final supernatants were then used for experimental analysis.

Metabolomic profiling was performed using ultra-high-performance liquid chromatography coupled with tandem mass spectrometry (UHPLC–MS/MS, Thermo Scientific, USA). Raw data were processed using XCMS for peak detection, alignment, and normalization (39). Metabolites were identified by matching against the Human Metabolome Database and the KEGG database. PCA was performed using MetaboAnalyst 5.0 to assess sample clustering, and orthogonal partial least squares discriminant analysis (OPLS-DA) was used to investigate metabolic variation patterns (40). Differentially expressed metabolites (DEMs) were identified based on variable importance in projection (VIP) scores, with thresholds set at VIP $> 1$ and $P < 0.05$ ($t$-test). Functional annotation and pathway enrichment analysis of DEMs were also conducted using the KEGG database (36).

## Integrated transcriptomic and metabolomic analysis

Based on our sample distribution in the OPLS-DA results, three metabolomics myce-lial samples were selected for integrated analysis with transcriptomics data. Pearson's correlation analysis was conducted to assess the relationships between DEGs and DEMs. The top 20 DEGs and DEMs were selected based on their absolute correlation coefficients. Subsequently, heatmaps were generated in R to visualize the association patterns between metabolites and genes. Following this, gene–metabolite interaction networks were constructed based on our KEGG pathway analysis to explore potential links between key regulatory genes and metabolic pathways to identify pathways co-enriched with DEGs and DEMs.

## Statistical analysis

All experiments were conducted with at least three independent biological replicates, and data are presented as mean ± standard deviation (SD). Biochemical data were analyzed using SPSS 27.0 (IBM, USA). Intergroup differences were assessed by one-way analysis of variance followed by Tukey's HSD post hoc test. $P < 0.05$ was considered statistically significant. Figures were created using Origin 2024 (Origin Lab, USA).

## RESULTS

### Antifungal activity of IVA, EMC, and IVA-EMC on *P. destructans*

VOCs inhibited *P. destructans* mycelial growth in a clear dose-dependent manner (Fig. 1A and B). The diameter of mycelial growth decreased progressively as VOC concentrations increased. Individually, IVA and EMC exhibited $IC_{50}$ values of 0.4 μL/mL and 1.52 μL/mL and MIC values of 0.64 μL/mL and 2.08 μL/mL, respectively. In fixed-ratio combina-tion assays, the dose–response curves shifted leftward, indicating increased antifungal potency; the $IC_{50}$ and MIC were 0.13 μL/mL IVA + 0.80 μL/mL EMC and 0.24 μL/mL IVA + 0.96 μL/mL EMC, respectively (Fig. S1A). Representative colony morphologies at the $IC_{50}$ and MIC are shown in Fig. 1C; Fig. S1B.

To rigorously quantify combination effects, we performed a vapor-phase checker-board assay and calculated the FICI using MIC endpoints (Fig. 1D). In the $6 \times 6$ matrix, two combinations achieved complete inhibition (FICI ≤ 0.5). The lowest FICI (0.375) occurred at 0.16 μL/mL IVA with 0.26 μL/mL EMC, delineating a ratio-dependent synergy window and indicating synergism between IVA and EMC. Outside this window, most combina-tions were additive, whereas few were indifferent at high or imbalanced doses (Table S2). Collectively, IVA and EMC effectively inhibit *P. destructans* growth, and their combination exhibits synergy within a defined dose–ratio window.

### Effects of IVA, EMC, and IVA-EMC on *P. destructans* cell wall and membrane integrity

As shown in Fig. 2A and B, treatment with VOCs at their $IC_{50}$ concentration caused notable alterations in the cell wall and cell membrane of *P. destructans* mycelia. In the control group, CFW staining showed uniform blue fluorescence in mycelia, while PI staining revealed no red fluorescence, which suggests intact cell walls and membranes. In the IVA and EMC groups, CFW staining showed uneven fluorescence in mycelia and reduced fluorescence at septa, which indicates cell wall damage. In contrast, the IVA–EMC combination group displayed CFW staining patterns similar to the control, implying minimal or no cell wall damage. All three VOC treatments showed strong red fluores-cence in PI staining that compromised membrane integrity.

To further evaluate membrane damage, MDA and ergosterol levels were quantified as key indicators. MDA levels in the control group were 8.4 nmoL/g, while those in the IVA, EMC, and IVA–EMC groups significantly increased to 14.5, 14.3, and 15.0 nmoL/g, respectively ($P < 0.05$), which is indicative of elevated lipid peroxidation (Fig. 2C). However, ergosterol levels decreased significantly in the IVA, EMC, and IVA–EMC groups

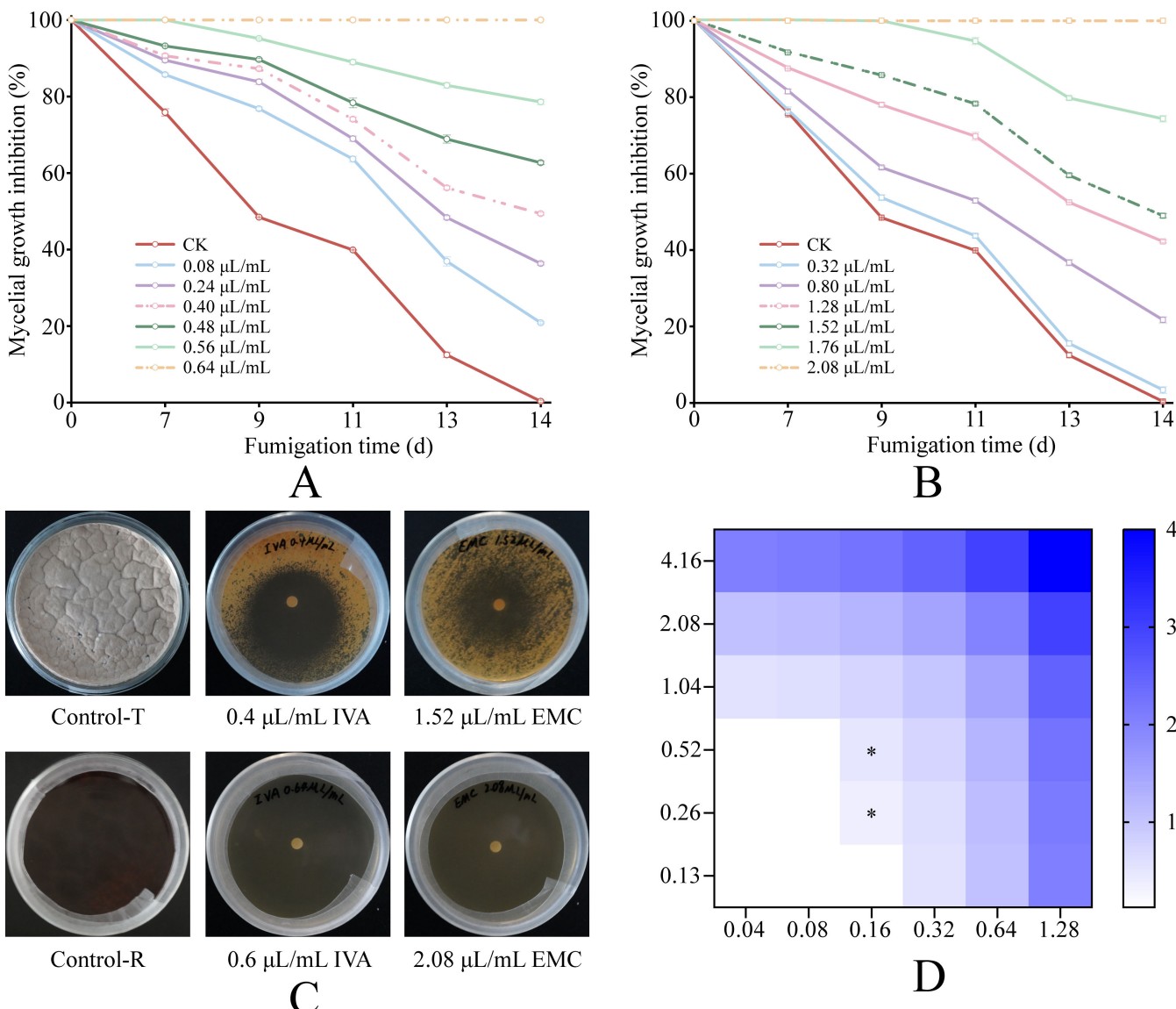

**FIG 1** Effects of VOCs on the mycelial growth of *P. destructans*. (A–B) Mycelial growth inhibition after 14 days of vapor-phase exposure to increasing concentrations of IVA (A) and EMC (B). (C) Representative colony morphologies under IC$_{50}$- and MIC-level exposures. (D) Vapor-phase checkerboard FICI heatmap for the IVA–EMC combination. Groups that reached the MIC endpoint are colored by the FICI category, while groups not reaching the MIC are left blank. Asterisks (*) indicate synergistic combinations (FICI ≤ 0.5). CK denotes the vehicle-only control. Points show mean ± SD (*n* = 3)

to 1.27%, 1.30%, and 1.00%, respectively, compared with 1.95% in the control (*P* < 0.05), which confirmed membrane damage (Fig. 2D). Overall, IVA and EMC disrupted both the cell wall and membrane, while the IVA-EMC combination affected only the membranes. Ultimately, all treatments increased MDA levels and reduced ergosterol content.

## Effects of IVA, EMC, and IVA-EMC on energy metabolism and oxidative stress in *P. destructans* cells

Compared to the control, intracellular ATP and NADPH levels increased significantly in treated mycelia (*P* < 0.05). Specifically, ATP levels rose 1.86-, 2.08-, and 2.74-fold in the IVA, EMC, and IVA-EMC groups, respectively (Fig. 2E), while NADPH levels increased to 2.1-, 2.54-, and 2.95-fold (Fig. 2F). Notably, the IVA-EMC combination produced the highest ATP and NADPH levels, which indicate the strongest impact on energy metabolism and oxidative stress. Furthermore, CLSM revealed remarkably increased ROS

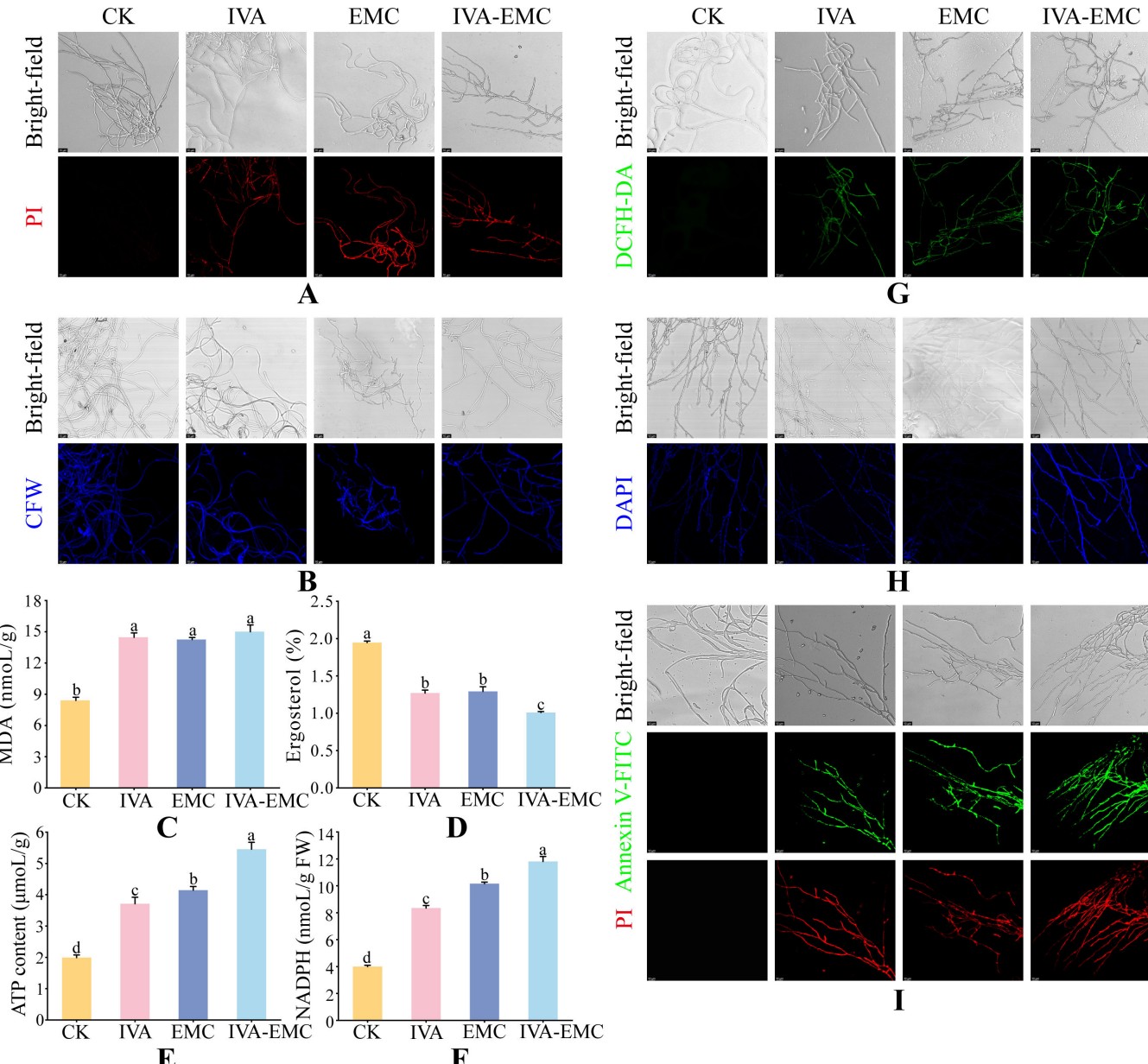

**FIG 2** Physiological and biochemical responses of *P. destructans* mycelia to IVA, EMC, and IVA–EMC treatments. CLSM fluorescence images of PI (A), CFW (B), DCFH-DA (G), DAPI (H), and Annexin V-FITC/PI (I) staining, respectively. Quantification of MDA (C), ergosterol (D), NADPH (E), and ATP (F) levels. CK denotes the vehicle-only control.

fluorescence in VOC-treated mycelia (Fig. 2G). Collectively, these results indicate that IVA, EMC, and IVA-EMC may inhibit *P. destructans* growth by interfering with energy metabolism and inducing oxidative stress.

## Effects of IVA, EMC, and IVA-EMC on DNA damage and apoptosis in *P. destructans* cells

DAPI staining revealed no noticeable nuclear morphological abnormalities in the IVA and EMC groups compared to the control. Mycelia displayed weak, uniform blue fluorescence, which suggests no significant DNA damage. In contrast, the IVA-EMC treated mycelia exhibited stronger fluorescence that indicated DNA damage and chromatin condensation (Fig. 2H). Furthermore, Annexin V-FITC/PI staining showed no detectable

Annexin V-FITC or PI fluorescence in the control group. In contrast, VOC-treated mycelia showed increased green and red fluorescence that implies increased apoptosis induced by VOCs in *P. destructans* (Fig. 2I). Interestingly, although IVA and EMC alone did not cause DNA damage, both still induced apoptosis. This suggests that apoptosis may be triggered through mitochondrial pathways, such as ROS accumulation (41).

## Effects of IVA, EMC, and IVA-EMC on transcriptomic profiles in *P. destructans*

To explore the response of *P. destructans* to IVA, EMC, and IVA-EMC, we performed RNA-seq to compare global gene expressions in treated and untreated mycelia. All samples had Q30 values > 95%, GC content > 54.49%, and over 94% of clean reads mapped to the reference genome (Table S2). PCA showed clear separation among treatments and tight clustering of replicates (Fig. 3A). A total of 12,274 DEGs were identified through comparative analysis. Notably, the IVA-EMC combination induced more DEGs (1,006 upregulated; 1,994 downregulated) than EMC (683 upregulated; 1,345 downregulated) or IVA (287 upregulated; 792 downregulated) alone, which suggests stronger gene regulatory effects (Fig. 3B). Additionally, 313 DEGs were commonly altered across all the treatment groups. To further explore the regulatory network, the top 15 hub genes were identified based on high connectivity (K ≥ 100) and edge weight (≥ 0.5). These included three transcription factors, five structural genes, and seven hypothetical proteins (Table S4), which were used to construct a regulatory network (Fig. 3C). Our functional annotation indicated that VOCs may affect metabolism, stress responses, cell growth, and protein degradation by modulating core transcription factors and metabolic genes.

KEGG enrichment analysis showed that IVA, EMC, and IVA-EMC significantly affected 11, 24, and 16 pathways, respectively (Table S6). Network analysis highlighted ribosome, glycine, serine, and threonine metabolism; oxidative phosphorylation; and glycolysis/gluconeogenesis as key metabolic pathways (Fig. 3D through F). Shared DEGs among all three treatments were significantly enriched in 10 pathways (Table S7), in which ribosome and glycine, serine, and threonine metabolism were highly represented (Fig. 3G and H). In summary, these VOCs likely inhibit *P. destructans* growth by interfering with protein synthesis, energy, and amino acid metabolism, with the IVA–EMC combination exerting the strongest transcriptional effects.

## RT-qPCR validation of key genes

To validate our transcriptome data, 12 key genes involved in chitin synthesis (*CHS1*), virulence secretion (*SP2* and *CRZ1*), energy metabolism (*ZWF1, SDH1,* and *ATP5*), purine metabolism (*PDE2* and *ADE13*), DNA replication (*MCM6*), cAMP signaling pathway (*PKA1*), and heat shock response (*HSP98* and *HSP90*) were selected for RT-qPCR validation. These RT-qPCR results were consistent with our RNA-seq data, which further confirm the reliability of our transcriptomic analysis (Fig. S2).

## Effects of IVA, EMC, and IVA-EMC on metabolomic profiles in *P. destructans*

To investigate how IVA, EMC, and the combined IVA-EMC treatments affect the metabolic network of *P. destructans*, we conducted metabolomic analyses using UHPLC-MS/MS. PLS-DA and OPLS-DA demonstrated distinct separation between the treatment groups and the control (CK), which suggests that exposure to these compounds induced significant biochemical changes in *P. destructans* (Fig. S3A through F). A total of 2,163 DEMs were identified across all the treatment groups. Compared to the CK, treatments with IVA, EMC, and IVA-EMC significantly altered 431 (215 upregulated; 216 downregulated), 517 (231 upregulated; 286 downregulated), and 452 (274 upregulated; 178 downregulated) DEMs, respectively. Notably, 97 DEMs were consistently affected across all the groups (Fig. 4A). Together, these findings indicate that IVA, EMC, and their combination substantially modify the metabolic profile of *P. destructans*.

We then conducted KEGG pathway enrichment analysis to identify the biological pathways associated with DEMs in each treatment group. As illustrated in Fig. S3G

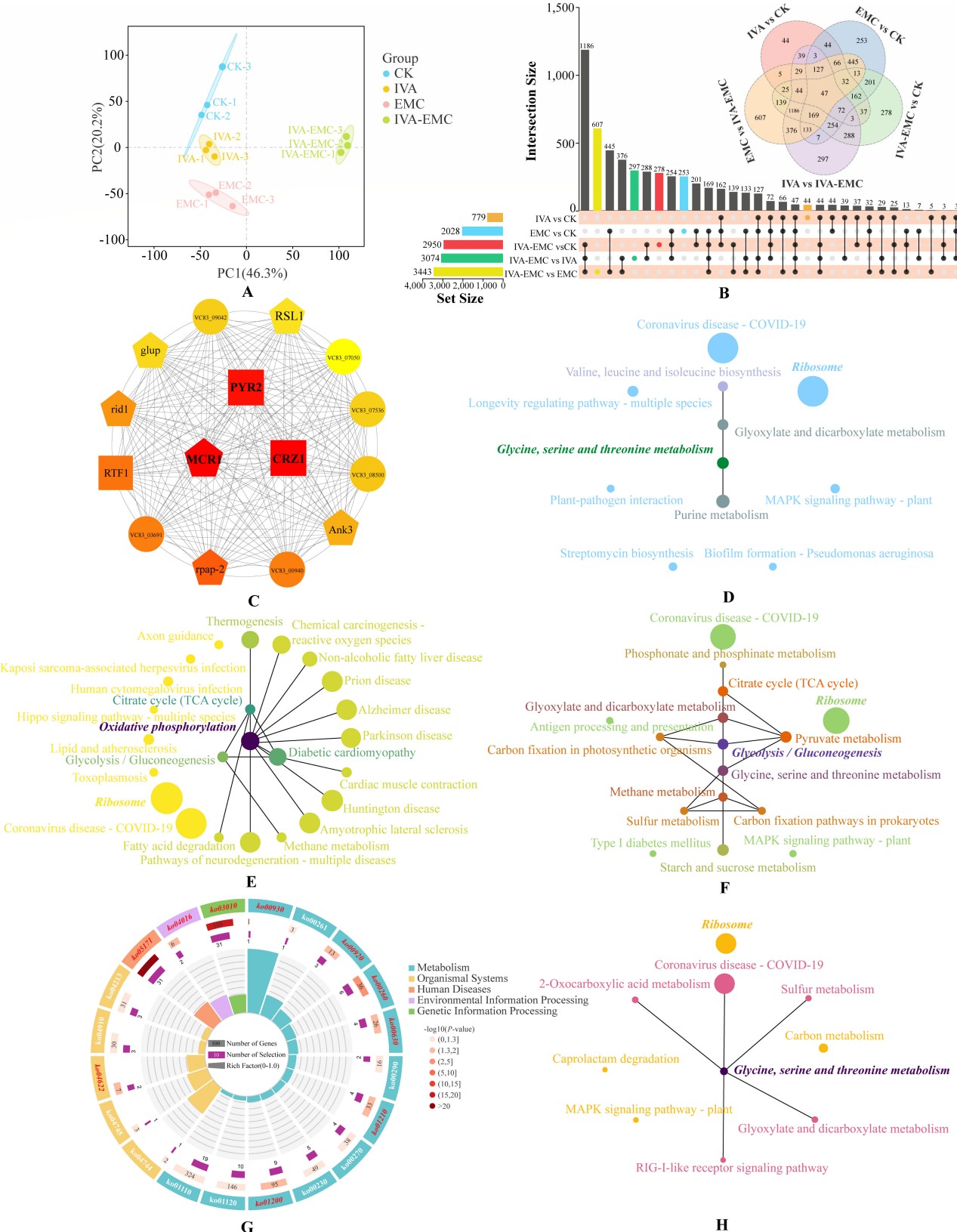

**FIG 3** Transcriptomic analysis of *P. destructans* mycelia under IVA, EMC, and IVA-EMC treatments. (A) PCA of gene expression profiles. (B) Upset plot shows DEG distribution across comparisons. (C) Correlation network of 15 core genes: transcription factors (squares), structural genes (pentagons), and unknown proteins (circles). Darker colors indicate higher ranks. (D–F) KEGG pathway network analysis for IVA (D), EMC (E), and IVA–EMC (F) groups. Node size denotes gene count, (Continued on next page)

Fig 3 (Continued)

and darker color indicates higher connectivity. (G) Circular plot of top 20 shared KEGG pathways. Significantly enriched pathways are highlighted in red italics. (H) Shared KEGG pathway network across treatments.

through I, the IVA, EMC, and IVA-EMC treatments significantly enriched 56, 45, and 50 pathways, respectively. Among these, 21 pathways were commonly enriched across the treatments. Notably, ABC transporters, biosynthesis of amino acids, and the cAMP signaling pathway were mainly induced (Fig. 4B). Additionally, the DEMs commonly affected by all three treatments were significantly enriched in critical metabolic pathways, such as ABC transporters (Fig. 4C). Detailed information on these enriched pathways is available in Tables S8 and S9.

## Integrated analysis of transcriptomic and metabolomic profiles

To further understand the antagonistic mechanisms of *P. destructans* in response to these compounds, we integrated transcriptomic and metabolomic data. As depicted in Fig. S4A through C, strong correlations were observed between the top 20 DEGs and DEMs, indicating a close association between gene expression changes and metabolite fluctuations (Fig. 4D). Furthermore, we mapped both treatment-specific and shared DEGs and DEMs to the KEGG database, which revealed several genes and metabolites were

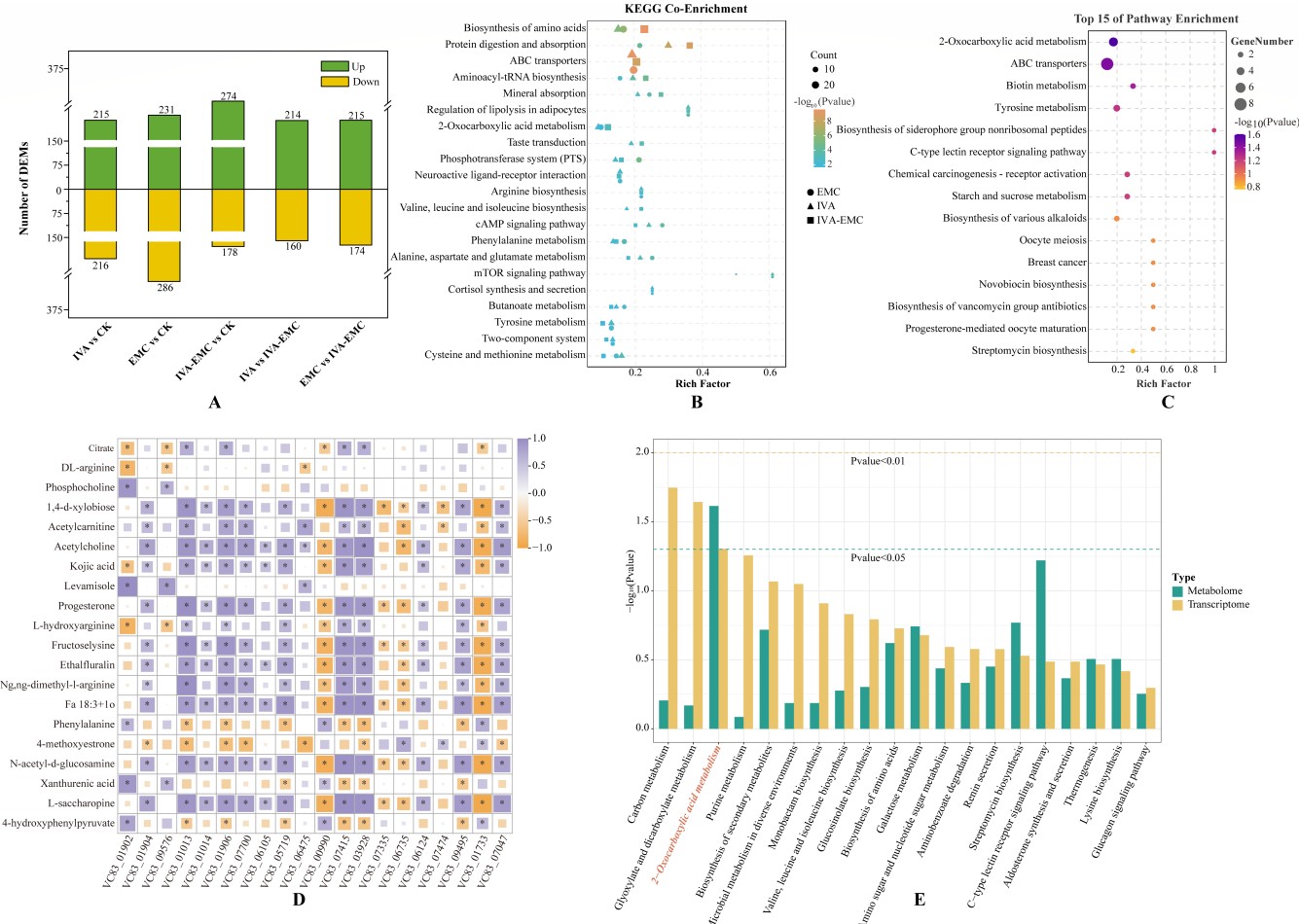

FIG 4 Metabolomic and integrative multi-omics analysis of *P. destructans* under VOC treatments. (A) Number of upregulated and downregulated DEMs in each comparison group. (B) Significantly enriched KEGG pathways shared by all treatment groups. (C) Top 15 KEGG-enriched pathways for the shared DEMs across the treatment groups. (D) Correlation analysis of the top 20 shared DEGs and DEMs across the treatment groups. (E) KEGG enrichment analysis of shared DEGs and DEMs across the treatment groups.

significantly enriched in the same pathways. This integrated analysis suggests that the toxicological effects of IVA, EMC, and IVA-EMC treatments may involve 4, 1, and 1 key functional pathways, respectively (Fig. S4D through F), and include critical metabolic routes like purine metabolism; glycine, serine and threonine metabolism; and the citrate cycle (TCA cycle) (Fig. 4E). Taken together, our findings indicate that these VOCs may inhibit fungal growth by disrupting intracellular pathways related to amino acid, purine, and energy metabolism.

## DISCUSSION

In recent years, several strategies have been proposed to combat *P. destructans*, including ultraviolet light (42), antifungal agents (43), probiotics (44), and environmental biocontrol agents (45). However, most of these strategies depend on contact-based inhibition and show limited effectiveness under natural environmental conditions. VOCs are low-molecular-weight chemicals with high volatility that are capable of diffusing through complex environments via air and typically exhibit low environmental persistence (46). Several studies have reported strong *in vitro* antifungal activity of various VOCs against *P. destructans*, which offer new prospects to control WNS (10). In this study, we report the *in vitro* inhibitory effects of two VOCs derived from bat cave environments in China, against *P. destructans*, individually and in combination. Together, our results provide new insights into their synergistic antifungal activity and the underlying molecular mechanisms.

The fungal cell wall is mainly composed of chitin, glucans, mannans, and glycoproteins that are essential to maintain cell morphology and environmental sensing. It also serves as a primary target for antifungal agents (47). In our work, CFW staining showed that both IVA and EMC treatments compromised the integrity of the *P. destructans* cell wall, whereas the combined treatment had no effect (Fig. 2F). This observation is consistent with an earlier study (23), which reported similar VOC-induced effects on *Botrytis cinerea* mediated by *Pseudomonas fluorescens* ZX. Our transcriptomic analysis also showed that IVA and EMC significantly downregulated the expressions of chitin synthase (CHS) I (CHS1, VC83_06323) and III (CHS7, VC83_05759), while the combined treatment upregulated CHS 2 genes (CHS2_1, VC83_02183 and CHS2_2, VC83_06199). Together, these results suggest that IVA and EMC may compromise cell wall integrity by downregulating CHS genes, whereas their combination may partially mitigate this effect by upregulating a subset of CHS transcripts.

The fungal cell membrane, together with the cell wall, maintains cell morphology and supports critical functions, such as transport and homeostasis, which makes it a key target for antifungal drugs (48). Our PI staining confirmed that all treatments disrupted the integrity of the *P. destructans* cell membrane (Fig. 2E). Additionally, metabolomic analysis further showed a significant decrease in glycerophospholipid-associated metabolites (e.g., glycerophosphocholine and choline) that may weaken membrane stability and permeability (49). Likewise, several membrane proteins related to ABC transporters were significantly affected, which is consistent with the antifungal mechanism of 2-phenylethanol against *B. cinerea* (15). Ergosterol, a major component of fungal membranes, contributes to membrane stability and fluidity and supports enzyme-membrane interactions (50). In this study, ergosterol levels were significantly reduced in all treatment groups (Fig. 2B), which is consistent with the downregulation of biosynthesis-related genes, like *ERG28* (VC83_01482) and *ERG8* (VC83_06822). Similarly, linalool disrupts membrane integrity in *B. cinerea* by downregulating ergosterol biosynthesis genes (51). Moreover, MDA, a key product of lipid peroxidation, is widely used as an indicator for oxidative membrane damage (52). Our results showed elevated MDA levels in all treatment groups, with the highest concentration in the IVA-EMC group (15 nmoL/g). This further confirms membrane damage caused by lipid peroxidation (Fig. 2A). Ultimately, these findings suggest that these compounds disrupt the structure and function of the *P. destructans* cell membrane by altering lipid metabolism, inhibiting ergosterol synthesis, and inducing lipid peroxidation.

As the causative agent of WNS, *P. destructans* has attracted considerable attention due to its diverse virulence traits, such as immune evasion, tissue invasion, nutrient acquisition, and stress adaptation (16, 17). During infection, *P. destructans* remodels its cell wall to evade, or in some cases hyperactivate, host immune responses (17). Transcriptomic analysis revealed significant changes in the expressions of genes involved in cell wall remodeling after VOC treatments, suggesting that these compounds may interfere with immune evasion by disrupting this process (Table S5). In addition, the 15 key genes we identified from the co-expression network were also strongly linked to virulence (Table S4). For example, the hub transcription factor RYP2 is essential for spore production in *Histoplasma capsulatum* (53), while CRZ1 plays a vital role in the pathogenicity of various plant and animal fungal pathogens (54). Furthermore, tissue invasion, a hallmark of WNS pathology (55), is closely associated with the expression of secreted proteases. We found that the subtilase family gene, *Pdsp1* (VC83_06062), was significantly downregulated by IVA and EMC treatments, while *Pdsp2* (VC83_04892) was downregulated in the EMC group. In contrast, the combination treatment did not affect these collagen-degrading enzymes (56). Metabolomic analysis also showed that all treatments significantly reduced riboflavin levels in the fungal mycelia. Previously, riboflavin accumulation in host skin has been linked to worse infection and tissue necrosis (57). Taken together, these observations support a potential attenuation of virulence-associated processes in *P. destructans*. However, we did not directly measure virulence or infectivity, and dedicated *in vitro* or *in vivo* assays will be required to test this hypothesis.

Energy metabolism is essential for fungal growth and pathogenesis, and its disruption can inhibit mycelial proliferation and reduce virulence (58). In this study, treatment with VOCs significantly activated the energy metabolism network in *P. destructans*. Notably, key enzymes and metabolites involved in oxidative phosphorylation and the TCA cycle were upregulated. These findings imply that the fungus enhances energy metabolism to counter environmental stress (Fig. 5), which is consistent with previous data that show human lactoferrin treatment increases cellular ATP levels in *Candida albicans* (59). During glycolysis, the expressions of several key enzymes increased and were accompanied by glucose consumption and pyruvate accumulation that indicate an enhanced metabolic flux toward the TCA cycle (60). Simultaneously, enhanced lipolysis resulted in fatty acid accumulation, which contributes acetyl-CoA to the TCA cycle. Increased amino acid metabolism also provided key intermediates, such as fumarate. The enhanced TCA cycle activity promoted the production of NADH and $FADH_2$, thereby accelerating oxidative phosphorylation and increasing ATP yield via ATP synthase upregulation (61). Our biochemical assays also showed that the treatments significantly elevated ATP levels in mycelia. Notably, the IVA-EMC combination group exhibited a 2.74-fold increase compared to the control, which surpassed the individual treatments and suggested a strong activation in energy metabolism (Fig. 2C). It is also important to note that elevated ATP levels may precede apoptosis, which is consistent with findings that *trans*-anethole induces elevated oxidative phosphorylation and ATP production that triggers apoptosis in *Aspergillus flavus* (13). However, mitochondria function not only as the central hub for energy production but also as a major source of ROS (62). Although multiple metabolic pathways collaboratively enhance ATP synthesis, excessive energy metabolism can lead to ROS accumulation and oxidative stress, which is an established antifungal mechanism of VOCs (15, 63). Our DCFH-DA staining revealed that VOC treatment significantly increased ROS accumulation in *P. destructans* mycelia, and these high ROS levels triggered the release of pro-apoptotic factors that led to apoptosis, which was confirmed by Annexin V-FITC/PI staining (Fig. 2G and I). Under oxidative stress, cells typically activate antioxidant systems to mitigate ROS damage. The oxidative phosphorylation pathway contributes NADPH that can maintain intracellular reducing power and supports antioxidant regeneration (64). In this work, VOC treatment significantly enhanced pentose phosphate pathway activity, primarily through the upregulation of glucose-6-phosphate 1-dehydrogenase (*ZWF1*, VC83_05048), thereby promoting NADPH

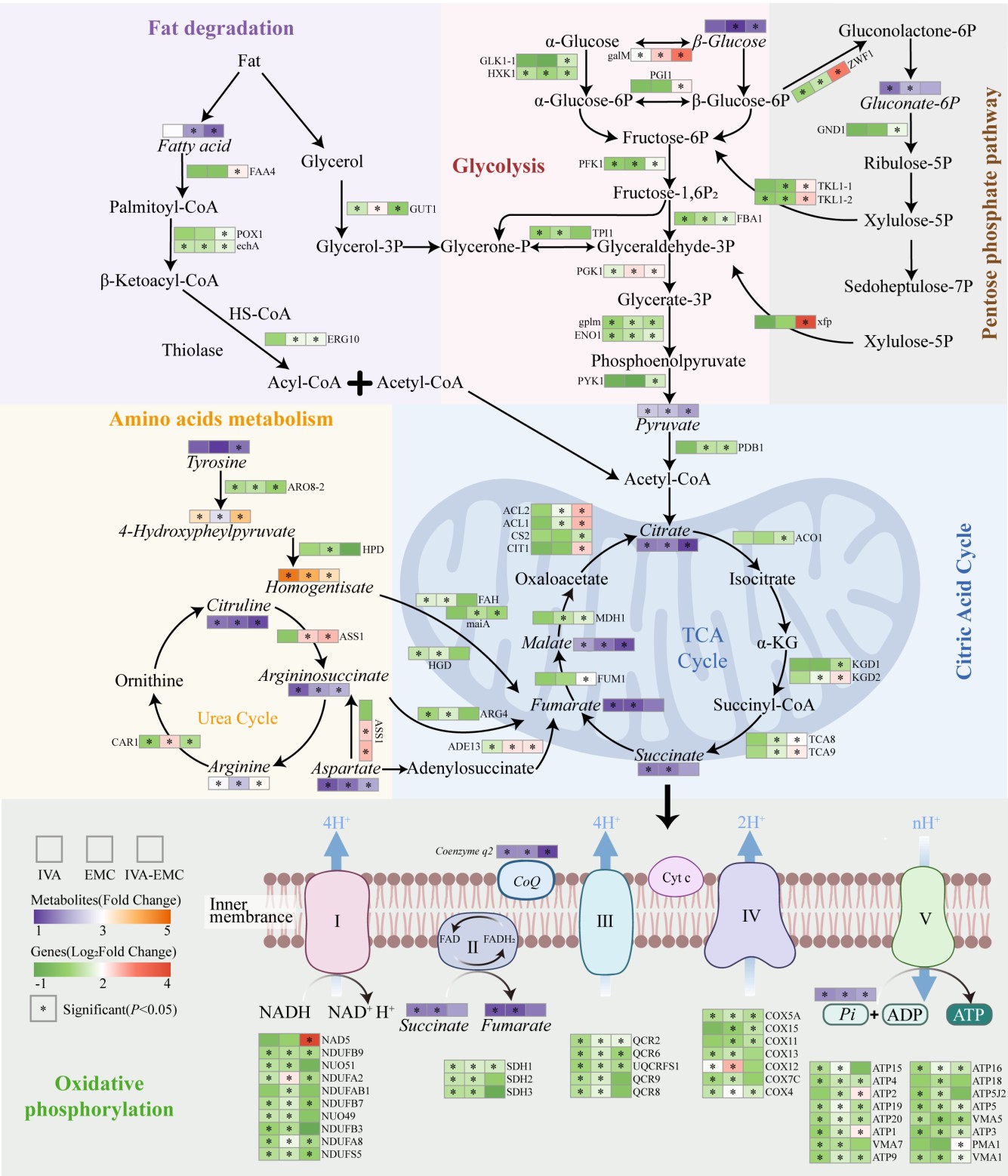

**FIG 5** Network representation of energy metabolism and related pathways in *P. destructans* under IVA, EMC, and IVA–EMC treatments. Italicized text distinguishes DEGs (*n* = 3) from DEMs (*n* = 6). Asterisks (*) indicate significant differences (*P* < 0.05). Abbreviations are shown in Table S10.

synthesis. Our biochemical assays also showed that NADPH levels in the IVA-EMC group were 2.95-fold higher than in the control and significantly exceeded those in the IVA

and EMC groups (Fig. 2D). Altogether, these data suggest that the fungus may enhance its reducing power to counteract oxidative stress. However, despite elevated NADPH levels, ROS accumulation persisted, which indicates that VOC-induced oxidative stress exceeded the antioxidant capacity of *P. destructans*. Ultimately, this redox imbalance may be a key mechanism underlying VOC-mediated fungal inhibition.

Purine metabolism plays a critical role in nucleic acid (DNA and RNA) synthesis and energy homeostasis maintenance that broadly contributes to diverse biological processes (65). Previous studies have shown that fluconazole stress disrupts purine metabolism in *C. albicans* (66). In this study, we saw that VOC exposure to *P. destructans* upregulated key genes in the *de novo* purine synthesis pathway, including *ADE1*, *ADE13*, and *ADE17* (Fig. 6). Overexpression of these genes in *Saccharomyces cerevisiae* has been shown to enhance cell growth under various stress conditions (67). Concurrently, AICAR, AMP, and cAMP levels significantly increased. Downregulation of *PDE2* (cAMP phosphodiesterase) further enhanced cAMP accumulation and suggests that *P. destructans* may upregulate *de novo* purine synthesis to counteract VOC-induced stress. Although all treatments enhanced purine metabolism, only the IVA-EMC combination significantly upregulated DNA replication-related genes (e.g., helicases, polymerases, and PCNA), indicating additional effects on DNA expression (68). DAPI staining revealed that IVA-EMC treatment induced DNA damage in *P. destructans* mycelia that was accompanied by the significant downregulation of the DNA repair protein *RAD16* (VC83_02915).

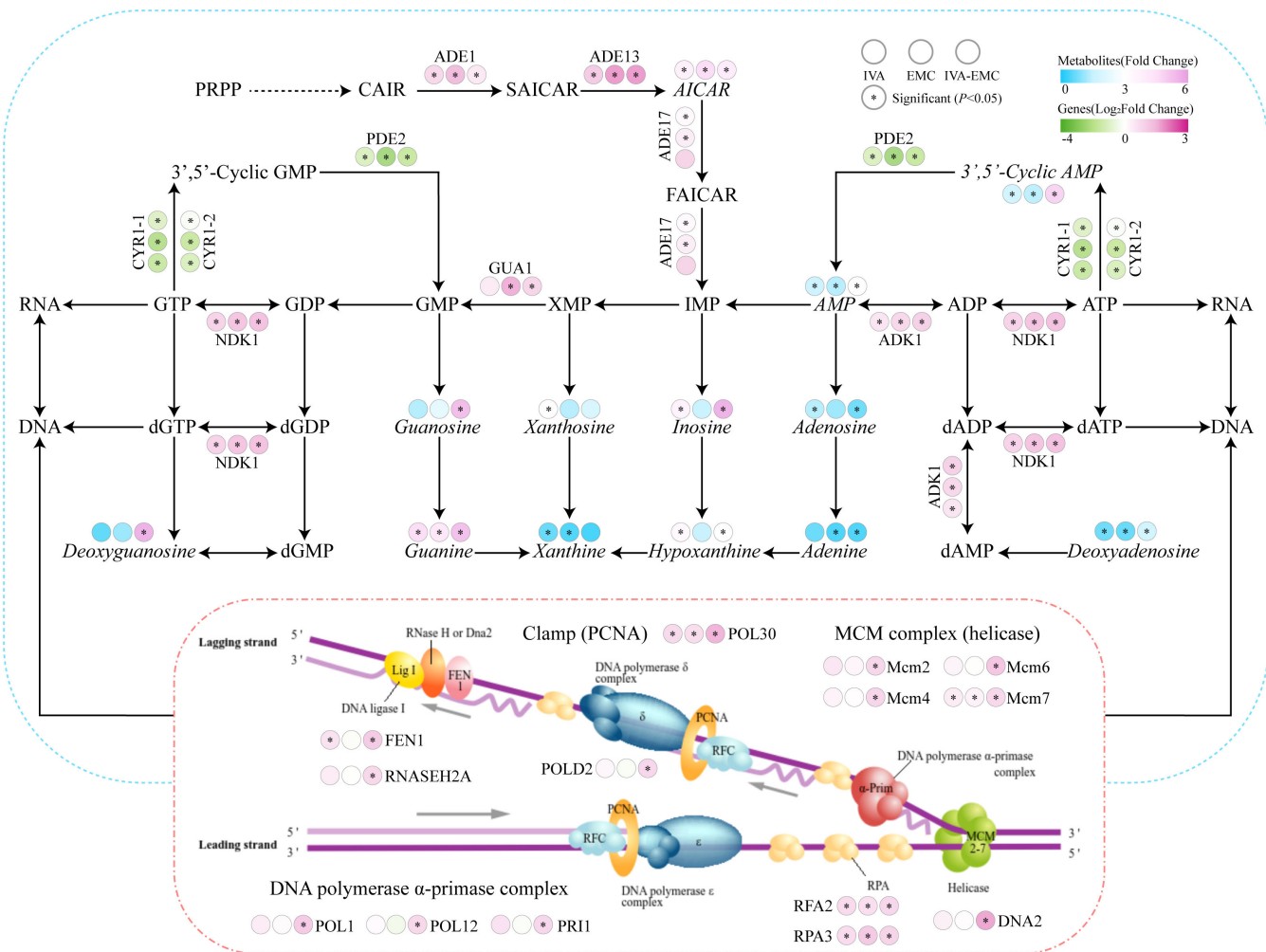

**FIG 6** Disruption of purine metabolism and its impact on DNA replication in *P. destructans* under VOC treatments. DEMs (*n* = 6) are italicized to distinguish them from DEGs (*n* = 3). Asterisks (*) indicate significant differences (*P* < 0.05). Abbreviations are shown in Table S10.

Impaired DNA repair, coupled with ongoing DNA replication, may increase genomic instability and mutation rates (Fig. 2H). Previously, Sun et al. reported that purine metabolism is closely linked to cell proliferation. However, DNA replication initiation also requires regulation by other factors, such as cell cycle signals (69). cAMP is a key signaling molecule in fungi that regulates cell growth, stress responses, and DNA damage repair (70). Under physiological conditions, cAMP activates downstream signaling via the protein kinase *PKA1*. In this study, only the IVA-EMC treatment significantly upregulated *PKA1* (VC83_05871), with a ~fivefold increase in cAMP levels, which indicates a strong activation of the cAMP signaling pathway (71). Ultimately, this process may contribute to aberrant DNA replication and damage accumulation. These findings indicate that IVA, EMC, and IVA-EMC treatments disrupt purine metabolism in *P. destructans*. The combined IVA-EMC treatment had the most pronounced effect that significantly activated DNA replication and the cAMP signaling pathway, while also reducing DNA repair capacity and exacerbating DNA damage. These results suggest that the IVA-EMC combination exerts synergistic antifungal effects.

In fungi, the cAMP signaling pathway regulates not only cell growth but also apoptosis and heat shock protein (HSP) expression. HSPs are considered both putative virulence factors and potential antifungal drug targets in *P. destructans* (17, 72). Many cAMP signaling proteins are HSP clients, and activation of the cAMP pathway typically suppresses HSP expression. Previous studies have found that trans-anethole treatment in *A. flavus* activates the cAMP signaling pathway and inhibits HSP function (13). In our work, we found that IVA-EMC treatment significantly downregulated several HSP-related genes, including *Hsp90* (VC83_08187), *Hsp70* (VC83_01046), and *Hsp98* (VC83_08137) (Table S5). Under stress, HSP expression mitigates cellular damage and promotes recovery, while its suppression may trigger apoptosis (73). We found that IVA-EMC treatment led to strong cAMP pathway activation, reduced HSP expression, and apoptosis induction in mycelia (Fig. 2I). Altogether, these findings suggest that IVA-EMC induces apoptosis by activating the cAMP pathway and suppressing HSP expression.

Although we used a vapor-phase exposure design—placing sterile antibiotic disks at the center of the Petri dish lid, opposite the medium—the inhibitory effects of IVA and EMC were not uniformly distributed (Fig. 1D). Maximal activity occurred adjacent to the disks. This pattern suggests that inhibition arises primarily from headspace concentration gradients, rather than from liquid diffusion into the agar. These gradients may be shaped by boundary-layer effects near the lid, incomplete mixing in a confined volume, differential adsorption to the agar surface, and vapor deposition onto the agar (74, 75). Moreover, the spatial distribution is compound-specific: under identical conditions, dimethyl disulfide produced near-uniform inhibition across the plate with no distinct inhibition zones, likely owing to its higher volatility and weaker surface affinity (76). Thus, physicochemical properties, together with the nominal dose, likely co-determine *in situ* performance and distribution. To clarify how vapor-phase transport governs inhibition, future work should quantify headspace concentrations and their temporal dynamics using closed-chamber sampling coupled to gas chromatography–mass spectrometry. The experimental geometry should also be optimized to promote uniform vapor exposure, for example, by using symmetric layouts, multiple low-load disks, or mixing elements.

For future applications, volatility confers clear advantages for interventions in hibernacula, including rapid dispersion, penetration into crevices, and the feasibility of noncontact treatment of bats (46). However, volatility also poses challenges, including dilution by ventilation, formation of spatial concentration gradients, and the need to maintain an effective concentration–time (C×t) exposure. In practice, experiments that mimic cave airflow and humidity should define target C×t metrics, select compounds and release modalities that sustain stable headspace concentrations, and preserve the ratio-dependent window identified *in vitro* when co-applying IVA and EMC. Because IVA and EMC are not risk-free, comprehensive assessment of nontarget effects is essential before deployment, including tolerance of bat respiratory and olfactory systems, impacts

on cave microbiomes and invertebrate communities, and environmental degradability. In summary, volatile compounds show substantial promise for WNS control, but effective implementation will require addressing these challenges through continued mechanistic studies and field validation.

## Conclusion

This work systematically investigated the inhibitory effects and molecular mechanisms of IVA, EMC, and their combination on *P. destructans*, using physiological assays, biochemical tests, transcriptomics, and metabolomics. Our results demonstrate that VOCs inhibit *P. destructans* growth via multiple mechanisms, including the disruption of the cell wall, cell membrane integrity, altered energy metabolism, oxidative stress and apoptosis induction, and modulation of pathways associated with pathogenicity. Additionally, we found interference with purine metabolism and the cAMP signaling pathway. The IVA–EMC combination exhibited ratio-dependent synergy in a vapor-phase checkerboard assay with MIC endpoints. Under effect-matched exposures, the combination exacerbated DNA replication stress and DNA damage and suppressed HSP responses, consistent with increased antifungal activity. In conclusion, this *in vitro* study provides new insights into VOC antifungal mechanisms and proposes potential strategies to control WNS in bats and other implications for ecological conservation.

## ACKNOWLEDGMENTS

This work was supported by the Jilin Provincial Natural Science Foundation (YDZJ202401501ZYTS) and National Natural Science Foundation of China (32300425, 32171525, 31961123001, and 32171481) and Innovation and Entrepreneurship Training Program for College Students in Jilin Province (202310193005 and S202410193089).

We also thank Dr. Jinhong Lei and M.S. Xuetong Liu for their help in data analysis and experiments.

Z.H.: Conceptualization, Methodology, Writing – original draft preparation, and Writing – review & editing. S.S.: Formal analysis and Validation. Y.L.: Software and Data curation. Z.W.: Writing – review & editing. M.S.: Validation. J.L.: Data curation. K.S.: Supervision, Project administration, and Funding acquisition. Z.L.: Supervision, Project administration, and Funding acquisition. J.F.: Supervision, Project administration, and Funding acquisition. All authors have read and approved the final work.

## AUTHOR AFFILIATIONS

[1]College of Life Science, Jilin Agricultural University, Changchun, China
[2]Jilin Provincial Key Laboratory of Animal Resource Conservation and Utilization, Northeast Normal University, Changchun, China
[3]Jilin Provincial International Cooperation Key Laboratory for Biological Control of Agricultural Pests, Changchun, China

## AUTHOR ORCIDs

Zihao Huang http://orcid.org/0009-0005-3407-992X
Jiaqi Lu https://orcid.org/0009-0009-9598-771X
Keping Sun https://orcid.org/0000-0002-4227-9818
Zhongle Li http://orcid.org/0000-0002-2137-9967
Jiang Feng http://orcid.org/0000-0002-7503-1069

## FUNDING

| Funder | Grant(s) | Author(s) |
| --- | --- | --- |
| Natural Science Foundation of Jilin Province | YDZJ202401501ZYTS | Zhongle Li |

| Funder | Grant(s) | Author(s) |
|---|---|---|
| National Natural Science Foundation of China | 32300425 | Zhongle Li |
| National Natural Science Foundation of China | 32171525, 31961123001, 32171481 | Keping Sun |
| Innovation and Entrepreneurship Training Program for College Students in Jilin Province | 202310193005, S202410193089 | Zhongle Li |

## AUTHOR CONTRIBUTIONS

Zihao Huang, Conceptualization, Methodology, Writing – original draft, Writing – review and editing | Shaopeng Sun, Formal analysis, Validation | Yihang Li, Data curation, Software | Zizhen Wei, Writing – review and editing | Mingqi Shen, Validation | Jiaqi Lu, Data curation | Keping Sun, Funding acquisition, Project administration, Supervision | Zhongle Li, Funding acquisition, Project administration, Supervision | Jiang Feng, Funding acquisition, Project administration, Supervision

## DATA AVAILABILITY

Transcriptome sequencing raw data were deposited in the NCBI SRA under accession number PRJNA1232437. Metabolome sequencing raw data are available at Figshare under DOI: 10.6084/m9.figshare.28551257. Supporting data for this study are available from Science Data Bank (DOI: 10.57760/sciencedb.22080).

## ADDITIONAL FILES

The following material is available online.

### Supplemental Material

**Supplemental Material (mSystems00903-25-s0001.docx).** Supplemental figures and tables.

### Open Peer Review

**PEER REVIEW HISTORY (review-history.pdf).** An accounting of the reviewer comments and feedback.

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
