## [Reviewer comments · mSystems]

Inhibitory and synergistic effects of volatile organic compounds from bat caves against *Pseudogymnoascus destructans in vitro*

Zihao Huang, Shaopeng Sun, Yihang Li, Zizhen Wei, Mingqi Shen, Jiaqi Lu, Keping Sun, Zhongle Li, and Jiang Feng

Corresponding Author(s): Zhongle Li, Jilin Agricultural University 吉林农业大学

Review Timeline:

Submission Date:	June 16, 2025
Editorial Decision:	September 13, 2025
Revision Received:	October 11, 2025
Editorial Decision:	November 21, 2025
Revision Received:	November 25, 2025
Editorial Decision:	December 2, 2025
Revision Received:	December 3, 2025
Accepted:	December 5, 2025

Editor: Yan Wang

Reviewer(s): Disclosure of reviewer identity is with reference to reviewer comments included in decision letter(s). The following individuals involved in review of your submission have agreed to reveal their identity: Christine E Salomon (Reviewer #2)

Transaction Report:

DOI: <https://doi.org/10.1128/mSystems.00903-25>

Re: mSystems00903-25 (**Inhibitory and synergistic effects of volatile organic compounds from bat caves against *Pseudogymnoascus destructans* in vitro**)

Dear Dr. Zhongle Li:

Both reviewers expressed interest in your work and recognized its value in this important field. They have provided detailed suggestions for improvement. Please ensure that your response letter and revised manuscript include the essential details, and that the methods are described in sufficient details to allow reproducibility. In addition, please consider increasing the font size in your figures to ensure readability.

Revision Guidelines

Sincerely,
Yan Wang
Editor
mSystems

Reviewer #1 (Comments for the Author):

This manuscript is clear and well done on an important topic. The work that has been performed was substantial and thorough. I

only have relatively minor suggestions.

MAJOR COMMENTS

None

MINOR COMMENTS

P2-L12. Add intro sentence about ubiquity of fungi and role of environmental reservoirs

P2-L12. Change "store" to "contain"

P2-L22. ROS acronym and others (e.g. MDA) not needed in Abstract if not used elsewhere in Abstract

P3-L44. Delete "the" before the word psychophilic

P3-L48. Add that hibernacula are the source of infection for returning bats

P3-L52. Suggest adding text that this is likely due to a long coevolution between bats and the fungus

P4-L53. Delete the word "other" as it makes the sentence unclear.

P6-L106. Possibly discuss controls here as well

P13-L249. Why would the IVA-EMC combination affect only the membranes?

P15-L292. Why are words such as Ribosome and Glycine and Oxidative capitalized?

P17-L325 Change Table to Tables

P18-L349. No need to define the WNS acronym again

P18-L364. I'm not sure I follow. While I understand the terms, can you please clarify "upregulating alternative isoforms"?

P22-L438. Change this to these

P26-L519 The quality of this top image could be improved.

P27-L526. acronyms like CK need to be defined in the legend.

P28-Fig. 3E. How do you explain diabetic cardiomyopathy in a fungus?

P30-L547. Figure can likely get moved to Supplemental

P31-552. Figure can likely get moved to Supplemental

Reviewer #2 (see attachment)

This manuscript describes a series of experiments to test and characterize the antifungal properties of the volatile compounds ethyl methyl carbonate (EMC) and isovaleric acid (IVA) as well as combinations of these compounds against the fungal bat pathogen *Pseudogymnoascus destructans*. Given the high interest in developing new treatments or mitigation strategies for WNS, this is an important topic and represents a useful approach towards assessing antifungal products. The series of activity characterizations are logical and useful for determining the potential mechanisms of action for the tested compounds. Although the overall determination about the antifungal activity is sound, there are a few important issues with the manuscript and with some of the conclusions.

One area of confusion in the manuscript is the use of the term “environmental reservoir”, which generally refers to a pathogen harboring location. In this paper, it seems to refer to locations that have VOCs present (presumably produced by microbes, although this is not clear). This could use some clarification throughout the introduction.

One of the important conclusions reached by the authors is that the combination of the two VOC compounds is “synergistic”, but this is not supported by any standard measurement or calculation, beyond just noting that the activity is greater using the combination at lower concentrations versus the individual compounds. A more rigorous determination would involve determining the fractional inhibitory concentration index and checkerboard assay, for example.

Similarly, it seems problematic to say that “the compounds reduce the virulence of *P. destructans* through multiple mechanisms that limit its ability to infect host tissues” when the actual virulence or infectivity of the treated fungus has not been tested in any in vitro or in vivo experimental systems. Even the incorporation of the words “may”, “suggest” or “support” would appropriately temper this conclusion.

One additional component that would be worthwhile to address in the discussion is the actual volatility of the tested compounds and limitations or advantages to this characteristic for these compounds. The disk diffusion assays suggest that there is some aqueous diffusion into the agar that is leading to the growth inhibition that is not consistent with only a volatile exposure effect (due to the inhibition not appearing uniformly across the plate and only adjacent to the disks). Since these compounds are being studied as potential treatments for mitigating *P. destructans* in caves, it is relevant to consider how they might be applied, and the relative concentrations needed for a killing exposure. Another component that would be useful to provide as context is any information known about potential toxicity or off target effects of these compounds.

There are a number of important details missing from the methods section.

Materials: Was *P. destructans* stored at -80 as spore stocks in water, or as glycerol stocks or as mycelial preparations?

Which ROS detection kit was used? (the dye manufacturer is mentioned but not an actual kit source)

What is meant by “activation of P destructans strain”?

The section on testing for antifungal activity using a disk diffusion assay is missing important details and is not reproducible in the current form. It is not clear how the pathogen was applied to plates. Although this states that compounds on disks were placed “opposite the fungal inoculum” on the plates, it is not clear if the inoculum was a spore preparation spread across the plate, or added as a mycelial plug, or some other way. Although compounds were added to disks at varying concentrations, it is not clear how this was done. What solvent were compounds diluted in? Although the results refer to specific concentrations, do these indicate the final concentration of material on the disk with respect to the volume of the agar? Or do they refer to the concentration of the solution added to the disk, and if so, what volume was applied? It would be more straightforward to provide the mass per disk, or to at least clarify that reference volume is the agar plate. The methods for quantifying inhibition are also not clear, since they involve the comparing growth diameters, but the experimental setup does not seem to conducive to growth diameter measurements (although the zone of inhibition could be measured and perhaps that is what is meant here). For the results of these experiments, the statement about “mycelial growth was completely inhibited at the MIC level....” is redundant, since this is the definition of an MIC/EC50, which were described above that sentence.

For the cell wall studies, additional details regarding the staining should be provided, such as the concentration and volume of dyes, solvents, etc. Similarly, the approximate amounts of mycelia used in the various experiments should be provided, such as for ergosterol and MDA determinations. These kinds of details should be provided for all of the methods sections in a way that would make the experiments reproducible by others.

Minor typos:

Page 3, line 44 “WNS is a psychrophilic fungal infection..” (remove extra “the”)

The following modified lines are all based on Manuscript.docx file

Reviewer 1

This manuscript is clear and well done on an important topic. The work that has been performed was substantial and thorough. I only have relatively minor suggestions.

MAJOR COMMENTS

None

MINOR COMMENTS

P2-L12. Add intro sentence about ubiquity of fungi and role of environmental reservoirs

P2-L12. Change "store" to "contain"

P2-L22. ROS acronym and others (e.g. MDA) not needed in Abstract if not used elsewhere in Abstract

Response: Thank you for your valuable suggestions. All of the above comments pertain to the Abstract, and we have revised this section accordingly:

1. We added an opening sentence to emphasize the ubiquity of fungi and the ecological role of environmental reservoirs in disease dynamics (Revised manuscript, lines 12-14).

2. At line 14, we refined the wording by replacing “store” with the more accurate term “contain.”

3. We removed unnecessary abbreviations and wrote out several terms in full for clarity (Revised manuscript, line 25).

P3-L44. Delete "the" before the word psychrophilic

P3-L48. Add that hibernacula are the source of infection for returning bats

P3-L52. Suggest adding text that this is likely due to a long coevolution between bats and the fungus

P4-L53. Delete the word "other" as it makes the sentence unclear.

Response: Thank you for pointing out these issues.

1. We have deleted “the” before “psychrophilic” in the Introduction (Revised manuscript, line 47).

2. We revised the text in the Introduction to clarify that hibernacula serve as infection sources for bats returning to these sites each hibernation season (Revised manuscript, line 50-52).

3. We added a statement in the Introduction (lines 56-68) noting that bats and *P.*

destructans have undergone a long coevolutionary history in China. This additional context supports the view that such coevolution may underlie the reduced disease impact observed in Chinese bat populations, complementing the discussion of immune function and microbiota.

4. To avoid ambiguity, we removed one occurrence of the word “Other” (Revised manuscript, line 59).

P6-L106. Possibly discuss controls here as well

Response: Thank you very much for your suggestion. We have clarified the control setup in the Methods section (Revised manuscript, lines 120-121). In this study, a vehicle control (CK) was used, consisting of sterile antibiotic discs loaded with 95% ethanol at the maximum solvent volume used in the treatment groups. This approach represents standard practice in vapor-phase antifungal assays. We also consulted and followed several comparable studies (e.g., doi: 10.1371/journal.pone.0179770; 10.1128/aem.00693-24; 10.1111/1751-7915.13765), which used identical control designs.

P13-L249. Why would the IVA-EMC combination affect only the membranes?

Response: We appreciate the reviewer’s insightful comment. We have clarified the interpretation of our findings in the Results and Discussion sections and revised the wording to avoid the misunderstanding that the compounds act solely on the membrane. Our integrated evidence indicates that the IVA-EMC combination primarily targets the cell membrane, while the cell wall shows no obvious morphological disruption but exhibits possible compensatory remodeling.

1. Evidence for membrane disruption: Both biochemical and omics data support membrane impairment under the combined treatment, including increased membrane permeability, lipid peroxidation (elevated MDA levels), and altered ergosterol-associated indices. These findings consistently indicate that the membrane serves as the major site of damage (Revised manuscript, lines 280-295).

2. Cell wall phenotype and compensatory response: CFW staining revealed no clear morphological damage to the cell wall under IVA-EMC treatment (Revised manuscript, line 282-283). However, transcriptomic analysis showed significant upregulation of chitin synthase genes (*CHS2_1*, *CHS2_2*) following the combined exposure, suggesting possible compensatory reinforcement of the cell wall. Accordingly, we replaced the previous general statement “upregulating alternative isoforms” with specific gene names and a more cautious phrasing (“may”) in the

revision (Revised manuscript, lines 404-406).

3. Supporting literature and mechanistic rationale: Consistent with our findings, Yue et al. (2023) reported that VOCs produced by *Pseudomonas fluorescens* ZX caused predominantly membrane damage with minimal effects on the cell wall in *Botrytis cinerea* (Revised manuscript, line 400). Given the hydrophobic nature of VOCs and the vapor-phase exposure design, preferential interaction with and disruption of the lipid bilayer is mechanistically plausible.

P15-L292. Why are words such as Ribosome and Glycine and Oxidative capitalized?

Response: We sincerely appreciate the reviewer's careful observation. In the original manuscript, we capitalized the initial letters of KEGG pathway names to maintain consistency with their official database format. However, following your suggestion, we have now standardized the capitalization throughout the text in accordance with general manuscript formatting conventions. The corrections have been made in the revised version (lines 335-337).

P17-L325 Change Table to Tables

Response: Thank you for pointing this out. We have corrected the usage to the plural form when referring to multiple tables in the revised manuscript (line 369).

P18-L349. No need to define the WNS acronym again

Response: Thank you for the helpful suggestion. We have removed the redundant definition of WNS in the revised manuscript (line 391). Upon further review, we also noticed a repeated definition of VOCs, which has now been similarly corrected at line 387 to streamline the text and avoid redundancy.

P18-L364. I'm not sure I follow. While I understand the terms, can you please clarify "upregulating alternative isoforms"?

Response: Thank you for requesting clarification. We acknowledge that the original wording was ambiguous. We have revised the statement to clearly explain that under the combined treatment, the fungus appears to compensate by upregulating alternative chitin synthase genes (CHS2 isoforms), which function as substitute enzymes for cell wall synthesis. By explicitly naming these isoforms and describing them as compensatory chitin synthases, we believe the revised text is now clearer and more precise (revised manuscript, line 406).

P22-L438. Change this to these

Response: We have revised the grammar as suggested. The sentence now uses

“these” instead of “this” to correctly refer to the plural subject, thereby improving grammatical accuracy and clarity (revised manuscript, line 480).

P26-L519 The quality of this top image could be improved.

Response: We have prepared a high-resolution version of Figure 1 (TIFF format, >300 dpi) and included it in the revised manuscript (line 591). All figures will be provided in high-resolution format to ensure optimal clarity and readability in both print and online versions.

P27-L526. acronyms like CK need to be defined in the legend.

Response: Thank you for highlighting this omission. We have revised the figure legend to define CK as the vehicle control group. In addition, we ensured that CK is clearly defined at its first occurrence in Figure 1 (line 598) and Figure 2 (line 603) so that readers can easily understand it without referring back to the main text.

P28-Fig. 3E. How do you explain diabetic cardiomyopathy in a fungus?

Response: Thank you for your insightful comment. In our study, KEGG enrichment analysis was performed using the full KEGG database (ko) rather than the fungus-specific subset. Pathways labeled with human disease names (e.g., diabetic cardiomyopathy, COVID-19, Parkinson disease) in KEGG actually represent conserved functional modules—such as oxidative phosphorylation, ROS generation and detoxification, and calcium homeostasis—that are shared across species. Therefore, their appearance in our fungal dataset does not imply that *P. destructans* possesses human-specific disease pathways, but rather that the underlying conserved mechanisms were enriched.

For instance, in Fig. 3E, the significantly enriched pathway Chemical carcinogenesis-reactive oxygen species involves conserved elements like mitochondrial electron transport and antioxidant systems, which are consistent with our biochemical ROS results. To avoid subjective bias or selective reporting, we did not manually remove these terms from enrichment figures. However, in Supplementary Tables S5–S8, we have clearly annotated each pathway’s higher-level KEGG category to help readers interpret their biological context.

Additionally, we have clarified in the Methods section (lines 211-214) that the enrichment analysis used the full KEGG (ko) database rather than a restricted fungal subset.

P30-L547. Figure can likely get moved to Supplemental

P31-552. Figure can likely get moved to Supplemental

Response: Thank you very much for this valuable suggestion. We have carefully reconsidered the possibility of moving the two mechanistic figures to the Supplementary Materials. However, we prefer to retain them in the main text because these figures are not decorative but serve as integrative summaries that link the transcriptomic and metabolomic datasets into a verifiable mechanistic framework (energy metabolism—glycolysis/TCA cycle—oxidative phosphorylation—ROS; and purine metabolism/cAMP signaling—DNA replication—HSP). They directly support the core arguments discussed in the Results and Discussion sections (revised manuscript, lines 447-526).

Presenting the multi-omics signals mapped directly onto pathway nodes within the main text allows readers to visualize the evidence-to-mechanism correspondence at a glance, improving interpretability and reproducibility while avoiding repeated cross-referencing between the main text and supplementary materials.

That said, we fully respect the reviewer's preference. If you still believe these figures would be better placed in the Supplementary Materials, we will make this adjustment in the next revision. We sincerely appreciate your thoughtful feedback, which has greatly improved the overall clarity and quality of our manuscript.

Thank you sincerely for all your comments!

Reviewer 2

This manuscript describes a series of experiments to test and characterize the antifungal properties of the volatile compounds ethyl methyl carbonate (EMC) and isovaleric acid (IVA) as well as combinations of these compounds against the fungal bat pathogen *Pseudogymnoascus destructans*. Given the high interest in developing new treatments or mitigation strategies for WNS, this is an important topic and represents a useful approach towards assessing antifungal products. The series of activity characterizations are logical and useful for determining the potential mechanisms of action for the tested compounds. Although the overall determination about the antifungal activity is sound, there are a few important issues with the manuscript and with some of the conclusions.

One area of confusion in the manuscript is the use of the term “environmental reservoir”, which generally refers to a pathogen harboring location. In this paper, it seems to refer to locations that have VOCs present (presumably produced by microbes, although this is not clear). This could use some clarification throughout the introduction.

Response: Thank you for this valuable suggestion. We have clarified and standardized the terminology in both the Abstract and Introduction to eliminate any ambiguity between environmental reservoirs and VOC sources. The revisions are as follows:

1. Abstract (lines 12-14): We now emphasize that environmental reservoirs can harbor pathogens, while VOCs are presented as potential components within the same environments, without altering the definition of reservoirs.

2. Introduction (lines 44-48): Based on prior definitions (e.g., doi: 10.1073/pnas.1914794117), we specify that environmental reservoirs refer to extra-host substrates (e.g., cave walls, sediments, and guano) that maintain *P. destructans* viability and serve as reinfection sources. We have inserted a concise definition at the first mention of the term and subsequently use hibernacula that harbor *P. destructans* or environmental reservoirs of *P. destructans* to avoid confusion with VOC-related contexts (revised line 51).

3. VOC sources: We have clarified that this study does not attempt to trace VOC origins. VOCs in cave environments may arise from microbial metabolism or from abiotic sources such as guano, substrate materials, or rock surfaces. To maintain neutrality, we consistently use the phrasing VOCs present in bat cave environments throughout the text. IVA and EMC were selected because both compounds were consistently detected at relatively high abundance across three hibernation sites in our prior GC-MS analyses, representing common VOCs in these environments. Regardless of their origins, their inhibitory effects on *P. destructans* were rigorously validated *in vitro* (revised lines 80-88).

One of the important conclusions reached by the authors is that the combination of the two VOC compounds is “synergistic”, but this is not supported by any standard measurement or calculation, beyond just noting that the activity is greater using the combination at lower concentrations versus the individual compounds. A more rigorous determination would involve determining the fractional inhibitory concentration index and checkerboard assay, for example.

Response: Thank you for your valuable suggestion. We fully agree that synergism should be determined using standardized methods and quantitative thresholds. Because the compounds in this study are volatile and act through vapor-phase exposure on solid media, we developed a gas-phase checkerboard assay compatible with this system, using the MIC as the endpoint and calculating FICI

values only for combinations that reached the MIC. The methodological details have been added to the revised manuscript (lines 128-145), and the corresponding data are provided in the Supplementary Materials. The specific revisions are as follows:

1. FICI calculation and design:

To maintain consistency with the main text, concentrations are reported as $\mu\text{L}/\text{mL}$; however, the FICI calculation uses fractional MIC ratios, and the units cancel out mathematically (revised line 116). The interpretive thresholds follow standard criteria: $\text{FICI} \leq 0.5$ indicates synergy; $>0.5-1$, additive; $1-4$, indifferent; and ≥ 4 , antagonistic (revised line 143). Based on single-compound MICs (IVA = $0.64 \mu\text{L}/\text{mL}$; EMC = $2.08 \mu\text{L}/\text{mL}$), we designed a 6×6 concentration matrix of $1/16\times$, $1/8\times$, $1/4\times$, $1/2\times$, $1\times$, and $2\times$ MIC levels. All assays were performed at 13°C and 85% relative humidity under sealed incubation for 14 days (revised line 137).

2. Clarification of fixed-ratio results:

To ensure accuracy, we corrected the statement regarding the fixed-ratio combination. The previously reported combination MIC ($0.24 \mu\text{L}/\text{mL}$ IVA + $0.96 \mu\text{L}/\text{mL}$ EMC) corresponds to $\text{FICI} = 0.24/0.64 + 0.96/2.08 = 0.84$, which falls within the additive range rather than synergistic. This has been revised in the main text (line 265), and the corresponding time–dose curve was moved to Figure S1 to avoid implying synergism from a single fixed-ratio shift.

3. Identification of a synergy window:

The two-dimensional checkerboard assay revealed a distinct synergy window, with two combinations showing $\text{FICI} \leq 0.5$ and the lowest value of $\text{FICI} = 0.375$. In Figure 1D, the full FICI heatmap visualizes synergistic (≤ 0.5), additive ($0.5-1$), and indifferent (> 1) regions, with non-MIC wells left uncolored. The apparent discrepancy between fixed-ratio and checkerboard outcomes reflects ratio-dependent interactions: a fixed ratio represents only one line through the dose–dose plane, whereas FICI mapping across two dimensions captures the geometry of vapor diffusion and concentration equilibration, thereby revealing synergism limited to specific ratio–dose combinations.

4. Consistency with mechanistic assays:

To prevent fungicidal effects from masking cellular responses, all mechanistic analyses (e.g., ROS, MDA, membrane/wall staining, omics) were conducted under subinhibitory, effect-matched exposures ($\sim 50\%$ inhibition). These assays are independent of the FICI-based synergism assessment. Specifically, combination

treatments used nominal concentrations corresponding to ~50% inhibition from the fixed-ratio experiment; these doses were explicitly stated as not used for synergy classification. Thus, the mechanistic data remain comparable to single treatments without affecting the rigor of synergy interpretation.

5. Terminology refinement for accuracy:

Although our new results demonstrate the presence of synergy, we revised the text to maintain cautious interpretation and avoid overgeneralization:

In the Abstract (line 20), “individual and synergistic effects” was changed to “individual and combined effects.”

In the Conclusion (line 563), we now describe the synergism more precisely and removed unnecessary linkage between omics findings and synergy per se, emphasizing instead the overall effects of co-treatment. The entire manuscript was checked to ensure that all related terms remain consistent and scientifically accurate. We sincerely appreciate your professional comments, which helped us greatly enhance the methodological transparency and overall rigor of our manuscript.

Similarly, it seems problematic to say that “the compounds reduce the virulence of *P. destructans* through multiple mechanisms that limit its ability to infect host tissues” when the actual virulence or infectivity of the treated fungus has not been tested in any *in vitro* or *in vivo* experimental systems. Even the incorporation of the words “may”, “suggest” or “support” would appropriately temper this conclusion.

Response: Thank you very much for this valuable suggestion. We agree that the current data are insufficient to directly conclude “reduced virulence.” Accordingly, we have revised the wording throughout the manuscript to emphasize that our findings indicate regulation or potential attenuation of pathogenesis-related pathways or processes, rather than direct evidence of virulence reduction. We also clarified that infectivity or pathogenicity was not directly measured *in vitro* or *in vivo*, and that these interpretations are based on indirect omics evidence requiring further validation. The specific revisions are as follows:

1. Abstract (line 27): the statement referring to “interference with virulence” has been revised to “regulation of virulence-associated pathways.”

2. Discussion (lines 444–446): we now specify that virulence processes may be attenuated, and explicitly note that this study did not directly assess infectivity or virulence, which will require future experimental verification.

3. Conclusion (line 561): the previous wording “reducing virulence” has been

replaced with the more precise phrasing “modulating pathways associated with pathogenic processes.”

These changes ensure the conclusions remain accurate and appropriately supported by the presented data. ◦

One additional component that would be worthwhile to address in the discussion is the actual volatility of the tested compounds and limitations or advantages to this characteristic for these compounds. The disk diffusion assays suggest that there is some aqueous diffusion into the agar that is leading to the growth inhibition that is not consistent with only a volatile exposure effect (due to the inhibition not appearing uniformly across the plate and only adjacent to the disks). Since these compounds are being studied as potential treatments for mitigating *P. destructans* in caves, it is relevant to consider how they might be applied, and the relative concentrations needed for a killing exposure. Another component that would be useful to provide as context is any information known about potential toxicity or off target effects of these compounds.

Response: Thank you for this constructive suggestion. We have expanded and streamlined the discussion to clarify the points you raised, summarized as follows:

1. Volatility and inhibition distribution (revised manuscript, lines 527-541): Since our assays involved vapor-phase exposure without direct contact between VOCs and spores, the strongest inhibition was consistently observed near the compound-loaded discs. This pattern likely reflects uneven vapor distribution within the sealed Petri dish rather than liquid diffusion into the agar. Notably, the spatial inhibition pattern varied by compound—some VOCs produced uniform inhibition across the plate, while others formed distinct inhibition gradients.

2. Field translation (lines 542-545): In future work, we plan to quantify the temporal dynamics of headspace VOC concentrations using sealed sampling and GC–MS analysis. This will allow conversion of the exposure endpoint into a concentration–time ($C \times t$, ppm \times h) metric and help optimize experimental geometry (e.g., symmetric or multi-disc low-load setups) to achieve more uniform vapor distribution.

3. Advantages, limitations, and safety (lines 546-554): VOCs offer key advantages such as rapid diffusion, penetration into microcrevices, and potential for non-contact treatment of bats. However, they also present constraints—ventilation can dilute vapors, spatial gradients may arise, and maintaining an effective $C \times t$ is essential.

Practical applications should therefore aim to replicate cave-like airflow and humidity conditions, set target C_{xt} values, and preserve the synergistic IVA–EMC ratio identified *in vitro*. Prior to field deployment, non-target effects (e.g., bat olfactory tolerance, impacts on cave microbiota and invertebrates, and compound persistence/degradation) must also be evaluated.

We believe these additions directly address your concerns and strengthen the bridge between our laboratory findings and potential field applications.

There are a number of important details missing from the methods section. Materials: Was *P. destructans* stored at -80 as spore stocks in water, or as glycerol stocks or as mycelial preparations?

Which ROS detection kit was used? (the dye manufacturer is mentioned but not an actual kit source) What is meant by “activation of *P. destructans* strain”?

Response: Thank you for your valuable comments. We have supplemented and standardized the relevant methodological information as follows:

1. Strain preservation and revival (revised manuscript, line 94):

We have specified that *P. destructans* was preserved as a 30% (v/v) glycerol spore suspension at -80°C . Before experiments, frozen stocks were thawed and streaked onto SDA plates, which were incubated at 13°C and 85% relative humidity for 14 days to obtain actively growing mycelia and fresh spore suspensions for subsequent assays.

2. Clarification of “activation of *P. destructans* strain” (line 104):

To avoid ambiguity, we have replaced “activation” with “revival” in both the subsection title and main text, and removed the word “strain.” In the opening sentence of this subsection, we clarified that “revival” refers to the thawing and subculturing of frozen spore stocks to obtain fresh viable spores.

3. ROS assay kit details and terminology standardization (lines 97–100):

We have added the full name and manufacturer of the ROS detection kit in the Materials and Chemicals section. Upon first mention in the text, it is now written as “Reactive Oxygen Species (ROS) Assay Kit (DCFH-DA).” The working concentration and incubation conditions have been provided in the methods. In addition, all abbreviations (CFW, PI, DCFH-DA, Annexin V–FITC/PI, etc.) are now defined consistently upon first use (lines 150, 152, 181, 192).

The section on testing for antifungal activity using a disk diffusion assay is missing important details and is not reproducible in the current form. It is not clear

how the pathogen was applied to plates. Although this states that compounds on disks were placed “opposite the fungal inoculum” on the plates, it is not clear if the inoculum was a spore preparation spread across the plate, or added as a mycelial plug, or some other way. Although compounds were added to disks at varying concentrations, it is not clear how this was done. What solvent were compounds diluted in? Although the results refer to specific concentrations, do these indicate the final concentration of material on the disk with respect to the volume of the agar? Or do they refer to the concentration of the solution added to the disk, and if so, what volume was applied? It would be more straightforward to provide the mass per disk, or to at least clarify that reference volume is the agar plate. The methods for quantifying inhibition are also not clear, since they involve the comparing growth diameters, but the experimental setup does not seem to conducive to growth diameter measurements (although the zone of inhibition could be measured and perhaps that is what is meant here). For the results of these experiments, the statement about “mycelial growth was completely inhibited at the MIC level....” is redundant, since this is the definition of an MIC/EC50, which were described above that sentence.

Response: Thank you for your helpful suggestions regarding methodological detail and reproducibility. We have revised and clarified the subsection “Revival of *P. destructans* and vapor-phase exposure to VOCs” accordingly. The main updates are as follows:

1. Inoculation method: We clarified that inoculation was performed using a spore suspension rather than mycelial plugs. Specifically, 100 μL of a 2×10^5 conidia/mL suspension was evenly spread onto the surface of 90-mm SDA plates and air-dried for 10–15 min before VOC exposure (revised manuscript, lines 105-111).

2. Application of compounds: We now specify that sterile antibiotic discs were fixed to the center of the inner Petri-dish lids directly opposite the agar surface. IVA and EMC were applied neat onto the discs at the designated loading volumes (lines 112-115).

3. Concentration definition and calculation: We clarified that the “nominal concentration” refers to the compound volume relative to the headspace air volume of the sealed plate ($\mu\text{L}/\text{mL}$). The formula is given as: $C_{\text{nom}} = V_{\text{voc}} / V_{\text{headspace}}$, where $V_{\text{headspace}}$ represents the volume of air above the agar. To ensure accuracy, headspace volume was determined experimentally by water-displacement, and the measured values are provided in Table S1 (line 116).

4. Control setup: Only vehicle controls (CK) were used, in which sterile discs were loaded with 95% ethanol at the maximum solvent volume used in treatments. This setup follows standard practice in vapor-phase antifungal assays. The CK data used for Figures 1–4 are derived from this vehicle control (lines 120-121).

5. Quantification and endpoints: Because inhibition in vapor-phase assays appears as growth suppression across the plate rather than discrete zones, we quantified inhibition as the mycelial coverage area (Ilastik + ImageJ). The inhibition rate was calculated as: $\text{Inhibition (\%)} = 1 - (A_t / A_c) \times 100$ (lines 122-124).

6. MIC definition: We removed redundant wording regarding “complete inhibition at MIC level” to improve clarity.

For the cell wall studies, additional details regarding the staining should be provided, such as the concentration and volume of dyes, solvents, etc. Similarly, the approximate amounts of mycelia used in the various experiments should be provided, such as for ergosterol and MDA determinations. These kinds of details should be provided for all of the methods sections in a way that would make the experiments reproducible by others.

Response: Thank you for your valuable suggestions regarding methodological details and reproducibility. We have revised and standardized the relevant sections of the Methods accordingly. The main updates are as follows:

1. Staining conditions: We have added detailed information on the working concentrations, volumes, solvents, incubation times, and temperatures for all staining assays, including CFW, PI, DCFH-DA, DAPI, and Annexin V–FITC/PI. The washing steps and number of rinses are now also specified. For clarity, PBS was used as the solvent unless otherwise noted; other staining buffers were provided within the respective commercial kits (their exact formulations are proprietary). (Revised manuscript, lines 150-152, 153, 180-182, 188-190, 191-194).

2. Sample mass: We now specify the mycelial mass used in each physiological and biochemical assay. For CFW, PI, ROS, DAPI, and Annexin V–FITC staining assays, approximately 20 mg wet mycelia were used per sample (lines 148, 181, 186). For MDA, ATP, NADPH, and ergosterol determinations, 0.1 g wet mycelia were used per assay (lines 158, 162, 177).

Minor typos: Page 3, line 44 “WNS is a psychrophilic fungal infection..” (remove extra “the”)

Response: We have corrected this grammatical error in line 47 of the revised

manuscript. Thank you for pointing this out. We sincerely appreciate all of your valuable comments, which have greatly improved the overall quality of our manuscript.

Thank you sincerely for all your comments!

Re: mSystems00903-25R1 (**Inhibitory and synergistic effects of volatile organic compounds from bat caves against *Pseudogymnoascus destructans* in vitro**)

Dear Dr. Zhongle Li:

Both reviewers are positive about your revised article. Please address the minor modifications suggested below.

Revision Guidelines

Sincerely,
Yan Wang
Editor
mSystems

Reviewer #1 (Comments for the Author):

L26. Remove the word "that" or finish the thought/sentence such as adding this phrase to the end of the sentence "are occurring".

Reviewer #2 (Comments for the Author):

Line 112: How were antibiotic disks affixed to the petri dish lids? (tape? Glue?)

The following modified lines are all based on Manuscript.docx file

Reviewer 1

L26. Remove the word "that" or finish the thought/sentence such as adding this phrase to the end of the sentence "are occurring".

Response: Thank you for pointing out this grammatical issue. We have revised the sentence accordingly. The text now reads:

“Transcriptomic and metabolomic analyses showed disruption of mycelial structure, modulation of virulence-associated pathways, induction of oxidative stress and apoptosis, and interference with purine metabolism, cAMP signaling, and energy metabolism.”

Thank you sincerely for all your comments!

Reviewer 2

Line 112: How were antibiotic disks affixed to the petri dish lids? (tape? Glue?)

Response: We appreciate the reviewer’s request for clarification. In our assays, the sterile antibiotic discs were not attached with tape or glue. Instead, the discs were placed on the inner surface of the Petri-dish lid, centrally over the inoculated agar surface. After the VOCs were loaded onto the discs, the plates were inverted and sealed with Parafilm so that the discs remained in a fixed position during incubation. To clarify this point, we have revised the Methods section (Lines 113, 119-121) as follows: “For vapor-phase assays, sterile antibiotic discs (6 mm; BKMAN, China) were placed on the inner surface of the Petri-dish lid, centrally over the inoculated agar surface.” “After VOC loading, plates were inverted, sealed with Parafilm, and incubated at 13°C and 85% relative humidity for 14 days, ensuring that the discs remained in position throughout incubation.”

Thank you sincerely for all your comments!

Re: mSystems00903-25R2 (**Inhibitory and synergistic effects of volatile organic compounds from bat caves against *Pseudogymnoascus destructans* in vitro**)

Dear Dr. Zhongle Li:

Thank you for submitting your revised work. Please ensure that Reviewer 2's query is explicitly addressed. Specifically: "Line 112: How were antibiotic disks affixed to the petri dish lids? (tape? Glue?)".

Revision Guidelines

Sincerely,
Yan Wang
Editor
mSystems

The following modified lines are all based on Manuscript.docx file

Reviewer 2

Line 112: How were antibiotic disks affixed to the petri dish lids? (tape? Glue?)

Response: We thank the editor and the reviewer for raising this important point. **We confirm that no tape, glue, or any other adhesive was used to fix the discs in place.** We deliberately avoided using any adhesive materials to prevent potential chemical interactions with the volatile organic compounds (VOCs) or sorption of VOCs by these materials, which could have affected the accuracy of the vapor-phase assay.

Our experimental design relied on gravity to maintain disc position. Specifically, after inoculation, Petri dishes were immediately inverted (i.e., agar surface facing upward and the lid facing downward). Sterile antibiotic discs were then placed centrally on the inner surface of the inverted lid (which in this configuration served as the “floor” of the headspace chamber). VOCs were loaded onto the discs at this position. Throughout incubation, the plates remained inverted and were sealed with Parafilm. Because the discs rested stably on the bottom of the sealed headspace chamber (the lid), they remained in place under gravity without any need for adhesives. This configuration has been consistently used in VOC-*P. destructans* assays (doi: 10.1128/aem.01187-25; 10.1080/21505594.2025.2569627; 10.1139/cjm-2020-0071).

We apologize for any confusion caused by the original wording regarding “inversion after loading.” We have revised the Methods section (Lines 112–114; 119–123) to clearly describe this physical orientation and fixation mechanism. The revised text now explicitly states that no adhesive materials were used and that the discs remained stationary solely by gravity throughout incubation.

Thank you sincerely for all your comments!

Re: mSystems00903-25R3 (**Inhibitory and synergistic effects of volatile organic compounds from bat caves against *Pseudogymnoascus destructans* in vitro**)

Dear Dr. Zhongle Li:

Your manuscript has been accepted, and I am forwarding it to the ASM production staff for publication. Your paper will first be checked to make sure all elements meet the technical requirements. ASM staff will contact you if anything needs to be revised before copyediting and production can begin. Otherwise, you will be notified when your proofs are ready to be viewed.

Sincerely,
Yan Wang
Editor
mSystems